

# Disturbances of Biological Soil Crust by fossorial birds increase plant diversity in a Peruvian desert

María Cristina Rengifo[1,3], Cesar Arana[1,2]

[1]Departamento de Ecología, Museo de Historia Natural de la Universidad Nacional Mayor de San Marcos, Lima, 15072, Perú.
[2]Laboratorio de Ecología y Biogeografía Terrestre de la Facultad de Ciencias Biológicas de la UNMSM, Lima, 15081, Perú.
[3]School of Forestry, Northern Arizona University, 200 E. Pine Knoll Drive, Flagstaff, AZ 86011, USA.

10    *Correspondence to*: Maria C. Rengifo (mcr335@nau.edu)

20




**Abstract.** The Lomas Formation are fog-dependent oases within the hyper arid band of the Peruvian coast. Biological soil crusts (BSC) form in the Lomas and interact with their fauna and flora. Here we asked if natural disturbances – biopedturbations - made by fossorial birds have an effect on seedlings emergence in the Lomas Formations in the National Reserve of Lachay in Lima, Peru. We analysed active and inactive avian biopedturbations, BSC and bare soil field samples for moisture content. Soil chemical properties were also analysed including organic matter, phosphorus, potassium, electrical conductivity (EC), pH and $CaCO_3$ content, from BSC, the soil beneath the BSC and soil from biopedturbations. Furthermore, we looked into the seedbank and the field emergence of seedlings in plots with BSC and with active and inactive biopedturbations. The results revealed that active biopedturbations had the highest soil moisture content and BSC showed the lowest values. Moreover, organic matter and potassium content were significantly higher in the BSC than the soil beneath it and the bare soil. On the other hand, $CaCO_3$ content and EC were higher in bare soil than the other treatments, and no significant differences were found in soil pH or phosphorus content between treatments. In the seedbank experiment, 13 herbaceous plant species were found; furthermore, biopedturbations had a higher diversity but lower abundance than the BSC. However, in the field observations biopedturbations had a higher diversity and abundance of seedlings than BSC and only 8 herbaceous species were found. The species *Fuertesimalva peruviana* (L.) Fryxell, *Exodeconus prostratus* (L'Hér.) Raf., *Cryptantha granulosa* I.M. Johnst. *Solanum phyllanthum* Cav. and *Calandrinia alba* (Ruiz & Pav.) D.C increase their abundance in some type of biopedturbations. Our results showed the positive effect on seed germination and diversity of vascular plants by the natural disturbances made by fossorial birds in a unique ecosystem of the Peruvian desert, and remarks the importance of spatial and temporal heterogeneity for ecosystem structure and functioning.



## 1 Introduction

Biological soil crusts (i.e., biocrusts or BSC) are important components of arid ecosystems (Johansen, 1993; Li, 2012; West, 1990). Crusts are initially formed by filamentous cyanobacteria, which produce exopolysaccharides that serve as agents for the amalgamation and stability of soil-particles (Johansen, 1993; West, 1990; Zaady et al., 2010). Early colonization by
cyanobacteria then facilitates the addition of other organisms that enrich the BSC community: green algae, lichens and mosses. The cryptogamic community increases the fertility of soils by fixing nitrogen and carbon, as well as affect the physical properties by altering water infiltration, runoff, albedo, and temperature (Belnap et al., 2016; Bowker et al., 2010; Prasse and Bornkamm, 2000; Weber et al., 2016; West, 1990; Zaady and Shachak, 1994). As a result, the emergence and survival of vascular plants is also promoted, making the BSC an ecosystem engineer (Jones et al., 1994, 1997).

The geographical location of the Sechura-Atacama Desert along the hyper-arid strip of the coast of Peru and Chile promotes the development of the Lomas Formation (Rundel et al., 1991). Despite the extreme climatic conditions, the coastal hills with Lomas communities are high in biodiversity and endemism of plants and animals (Dillon and Rundel, 1990; Ferreyra, 1983; Pulido et al., 2007). The dry conditions limit the establishment of perennial plants, allowing the development of annual vegetation only in the winter months when marine fog and fine drizzle provide water.

Several types of animals generate disturbances – biopedturbations – that break the BSC on the Lomas of the central Peruvian coast. For example, common birds such as miners *Geositta* spp. and the burrowing owl *Athene cunicularia* excavate tunnels and create mounds that litter the landscape. These disturbances occur mostly in the low-elevation areas of the hills (between 150 and 400 meters) where vegetation cover is limited and the soil cover is dominated by BSC (Ferreyra, 1953).

Previous studies have shown that biopedturbations from small mammals in arid and semiarid environments alter patterns soil
structure, nutrient availability, microtopography, and change the plant diversity (Eldridge et al., 2012; Jones et al., 1994; Kerley et al., 2004; Pickett and White, 1985; Whitford and Kay, 1999). Although fossorial behaviour has also been seen in some wide range birds, there has been very little mention of the presence and effect of birds as disturbance agents (McKechnie, 2006; Wilkinson et al., 2009) and little is known about its effects in natural environments. Animal burrowing redistributes resources for other species, which is the classic example of ecological engineering (Guo, 1996; Hansell, 1993;
Jones et al., 1994; Wright et al., 2004).

The present study aims to study the effects of BSC biopedturbations on the germination and community composition of herbaceous plants at the Lomas de Lachay natural reserve, Peru. We focus on the biopedturbations made by fossorial birds on the BSC. We tested the hypothesis that the herbaceous community increases their diversity in response to biopedturbations of the biological soil crust in the Lomas ecosystem. To asses our hypothesis we analyze soil moisture
content, soil chemical properties, and seedling emergency between biopedturbations and soil cover with BSC.

## 2 Methods

### 2.1 Study site

The National Reserve of Lachay is located 105 km north from the city of Lima, in the central coast of Peru (S11°23,6', W77°23'). The reserve contains a unique fog and mist-fed ecosystem called Lomas Formations within the hyper arid band of
the Peruvian coast (Fig. 1). The landscape is characterized by small hills that create a smooth gradient from 150 m to 750 m of altitude. In the high humidity season, from July until September, a dense fog comes from the sea and moisture the hills allowing the establishment of endemic flora (Rundel et al., 1991).

The study area is located in the lower part of the Lomas Formations, from 150 to 250 m of altitude. The seasonal vegetation is characterized for the presence of herbaceous plants of rapid flowering like the succulent *Cistanthe panicualata* (Ruiz &
Pav.) Carolin ex Hershk (Portulaccaceae), and the bulbous *Stenomesson coccineum* Herb. (Amaryllidaceae). The sandy loam soil is covered by a dark biological soil crust dominated by cyanobacteria (Arana et al., 2016), which is a type of BSC





commonly found in warm deserts (Belnap et al., 2001; Pietrasiak, 2014). The BSC is 1-5 mm thick, dominated by cyanobacteria but with a low percentage (less than 50%) of moss (*Bryum argenteum* Hedw mostly) and some crustose and fruticose lichens. Three species of fossorial birds are the main agents of biopedturbations on the landscape: the burrowing owl (*Athene cunicularia*), the coastal miner (*Geositta peruviana*) and the greyish miner (*Geositta maritima*). The
biopedturbations made by the fossorial birds creates a mound from the removed soil that lies on top of the BSC surface.

2.2 Moisture experimental design

We established 26 experimental sites, each of them consisting of three 30 x 30 cm experimental units: the bare soil plot (Experimental plot), the biopedturbation plot, and the BSC plot (an area with undisturbed BSC). Biopedturbation plot is the mound of sand made by the fossorial birds, and the bare soil plot is an area where we removed the layer of BSC. Selected
biopedturbations were separated at least 10 m. We considered two types of biopedturbations: active and inactive; and stablished 15 experimental units for active biopedturbations and 11 for inactive biopedturbations. In this manner, we look into the four types of possible soil profiles and their moisture content.

We sampled 100 g of the top 5 cm of soil in each plot, at three times: day 0, 5 and 60. At day 0 the bare soil plot was sample with the undisturbed BSC layer, and immediately after the collection of the sample the BSC was removed. Samples were
collected in hermetic bags, and later in the laboratory were weighted, dried in a stove for 24 h at 105 °C, reweighed and their volumetric moisture content calculated (Kidron and Tal, 2012; Yair et al., 2011).

2.3 Soil chemical properties

We sampled three soil treatments, each replicated three times and separated at by at least 10 m. Each replicate consisted of three subsamples separated by approx. 1 m. Every soil sample weighted 500 g approximately. There were three soil
treatments analysed: (1) undisturbed layer of BSC, (2) underlying soil 5 cm deep which was directly below the initially withdrawn BSC layer, and (3) removed soil from the loose sand of the entrance of the burrows. The analysis of the phosphorus (K), exchangeable potassium (K), calcium carbonate ($CaCO_3$), organic matter (O.M.), pH and electrical conductivity (E.C.), was made by the Laboratory of water, soil, environment and fertigation of the Universidad Nacional Agraria La Molina, in Lima.

2.4 Seedbank experiment

To evaluate the seed bank we took paired samples of a disturbed soil and an undisturbed BSC in field plots of 10x10 cm found next to each other by approx. 1 m. We used 27 active biopedturbations and 34 inactive biopedturbations, each paired with undisturbed BSC sample. The soil samples were taken from the top 5 cm of soil.

We conducted the experiment in a greenhouse at the UNMSM. The soil samples were placed in plastic trays and arranged in
a Latin Square design to eliminate positional effects within the greenhouse. The trays were watered regularly every two days with tap water, and the number of seedlings were recorded at frequent intervals. The germination beyond five weeks was found negligible and during the germination period there was no soil biopedturbation and seedling were removed only at the end of the experiment. The taxonomic determinations of species were made with dichotomous keys and specialized bibliography (Fryxell, 1996; Krapovickas, 2007; Lleellish et al., 2015; Sagástegui and Leiva, 1993; Tate, 2011).

2.5 Field seedlings emergence

We established a new set of experimental units, a30 x 30 cm plot on the sand mound of 30 biopedturbations that were separated by a minimum distance of 10m from each other. Next to each sand mound plot, separated by approx. 1 m, we placed another plot with the same dimensions in the undisturbed BSC. Paired sampling was needed to diminish the environmental heterogeneity and compare plots with similar conditions due to their proximity. We consider 15 replicas for



each type of biopedturbation (active and inactive). In every plot we count and identify the all present seedlings. We evaluate the seedlings emergence at the beginning of the wet season in August 2016.

## 3 Results

### 3.1 Soil moisture content

Active soil biopedturbations had the highest moisture content among the soil treatments at day 60 (Fig. 2). Inactive biopedturbation had similar moisture content to bare soil, and BSC had the lowest moisture content of all treatments (Fig. 2). Comparisons between types of biopedturbations was statistical significant (Median test, $p = 0.021$), active biopedturbations showed the highest moisture content, time was a negligible factor.

### 3.2 Chemical properties

Our results show that biopedturbations made by fossorial birds did not change the nutrient content of the soil (Table 1). In addition, organic matter showed higher values in the BSC, but electric conductivity and calcium carbonate were higher in the removed soil ($p < 0.05$, for both cases).

### 3.3 Effects on vegetation

    The seedbank experiment showed the germination of 13 plant species; where the BSC samples had a higher abundance of
germinated seeds for active and inactive biopedturbations. In the rarefaction curves the number of species expected was higher for the inactive bioperturbation plot in contrast with the paired BSC, followed by the active bioperturbation and the paired BSC.

    In the field we recorded the emergence of a total of eight species. Both the abundance and the diversity of plants were higher in both types of biopedturbation plots rather than in the paired BSC. Diversity index values didn't show statistical
significance, although inactive biopedturbations plots had a tendency of a higher diversity index compared to the paired BSC (Wilcoxon test, $p=0.507$). In the rarefaction curves the number of expected species was higher in the inactive biopedturbations, followed by the active biopedturbations, and the paired BSC had the lowest values (Fig. 3).

    *Cistanthe paniculata* was the species with the higher density in both seedbank and field emergence experiments, but only in the field observations density values were higher in the active biopedturbation plots. *Fuertesimalva peruviana* had a greater
density in active biopedturbations, while *Exodeconus prostratus, Cryptantha granulosa, Solanum phyllanthum* and *Calandrinia alba* had a higher density in the inactive biopedturbations (Table 2).

## 4 Discussion

    The results of our study showed that although birds disturbed the BSC, it increased plant diversity. Plant community showed an effect from both active and inactive biopedturbations. We also found soil moisture content was reduced by BSC, as a
possible result of a low water infiltration, because the BSC tends to seal the soil surface (Brotherson and Ruthforth, 1983; Zhang et al., 2010), even though some studies didn't find differences between bare soil and BSC (Evans and Johansen, 1999).

    Interactions among precipitation, BSC characteristics, topography and soil types (Chamizo et al., 2012) can cause BSC to increase (Brotherson and Ruthforth, 1983; Bu et al., 2015) or decrease (Eldridge et al., 2000; Gao et al., 2010; Kidron and
Yair, 1997; Yair, 1990) infiltration rates in the soil. It was suggested that in sandy soils the BSC decrease the infiltration as a result from the reduction in the porosity (Warren, 2001). In this manner it was expected that the BSC in our study area decreased the infiltration and as a consequence diminish the soil moisture content, as it was reported in studies in the Negev desert that has physical characteristics similar to our studied area (Eldridge et al., 2000; Keck et al., 2016). Biological soil





crust will seal the soil surface (Booth, 1941; Brotherson and Ruthforth, 1983; George et al., 2003); whereas the cyanobacteria and lichen decrease the permeability of the soil (Loope and Gifford, 1972), the bare soil doesn't (Keck et al., 2016). The hydrophobicity of the BSC is control by the compounds exuded by cyanobacteria and other organisms of the BSC; and the pore obstruction produced by the swelling of the exopolysaccharides prevents the rapid hydration of the soil.

The biopedturbations made by fossorials birds consist in the overlap of sand on BSC layers. Whereas sand layers allow water to infiltrate and BSC doesn't, the profile of a biopedturbation (where a mound of sand is on top of the BSC) will allow better moisture retention and the BSC layer, few centimetres under, will stop the infiltration. In a season of poor humidity, as the one occurred in our sampling season, the biological soil crust consumes the water available in the most superficial soil layer, and in consequence decrease the soil moisture content (Gao et al., 2010; Bu et al., 2015). Under these conditions, the
differences observed in the soil moisture content between treatments are not large but are more relevant to vegetation. In summary, the bird's biopedturbations will generate an area with higher soil moisture content than the BSC around it, and this evidence their importance.

Chemical properties in the soil had different values between the treatments. Soil pH was not altered significantly by the presence of BSC, and has been reported in other studies (Evans and Johansen, 1999; Guo et al., 2008; Kidron et al., 2015).
Although birds dropping may increase the fertility in biopedturbations, the nutrients potassium and phosphorus didn't show statistical differences between treatments, by which we induce that bird's biopedturbations don't alter significantly the nutrient soil content. Nevertheless, the potassium content had a tendency to be higher in the BSC, and it has been established before that BSC increases the soil fertility (Belnap and Harper, 1995; Harper and Pendleton, 1993). Organic matter was expected to be higher in the BSC in comparison with the underlying and removed soil, since is directly related to the organic
carbon and BSC have high concentration of organism living in it (Delgado-Baquerizo et al., 2016; Guo et al., 2008). Unlike other studies (Guo et al., 2008), calcium carbonate content was not higher in the BSC instead the removed soil had the higher values, this results may be explained by the abundant remains of gastropods shell's present in the locality. At the same time, since electric conductivity is influenced by the concentration of calcium carbonate BSC did not show high values of EC; despite being expected due to the concentration of ions released by the decomposition of microorganisms.

Studies on BSC effects on vascular plants germination and establishment have contradictory results (Bowker, 2007) some consider that BSC can affect negatively plant density (Johansen, 1993; Prasse and Bornkamm, 2000), other showed the positive effect in semiarid ecosystems (Boeken et al., 2004; Defalco et al., 2001) and only in specific species (Su et al., 2009). We used the data of seedling germination as an estimate of the seedbank present in the soil taken as reference other studies (Thompson and Grime, 1979; Zhang et al., 2010). Our seedbank experiment showed that the soil of the
biopedturbations was significantly less abundant in seeds of annual vascular plants in contrast with the BSC; nevertheless, was more diverse and rich in species composition. Even though a smooth biocrust increases the chances that seed may be taken by the wind (Belnap, 2006), plant litter on the soil surface remained along with a large seed load. The biopedturbations studied are a mound of sand over the BSC that visibly do not present the same quantity of plant litter and as a consequence it would be expected to have less seeds. A higher diversity in the biopedturbation evidence a positive effect in the vegetation
composition, but it would be necessary to study more thoroughly this phenomenon to understand if those biopedturbations work as a seed tramp or if the higher diversity is product of the soil removal by fossorial birds.

On the other hand, field data taken in the transition from dry season to wet season showed that the biopedturbation had higher abundance and diversity in contrast with the BSC. Moreover, biopedturbations had a positive effect in some species: *Fuertesimalva peruviana,* in spite of been a high density specie in the BSC, duplicated its abundance in active perturbations;
*Exodeconus prostratus* the species with the lowest density was only found in active biopedturbations; *Cryptantha limensis*, *Solanum phyllantum* and *Calandrinia alba* showed a higher density when emerged from inactive biopedturbations. We hypothesized that some species benefit from the higher moisture content in the biopedturbations, and some may also be




favoured by other characteristics of the biopedturbations. Further studies would be necessary to get a better conclusion. Nevertheless, we evidence the positive effect on specific species as the literature mentioned it (Su et al., 2009).

The three components of the ecosystem studied have a net of interactions, positives as negatives, and our work focus on part of them. The biopedturbations made by fossorial birds, meaning the destruction of the BSC, creates an area where the biological soil crust is buried under a mound of sand and this burial causes a negative effect to the BSC (Rao et al., 2012). The interaction between biological soil crust and vegetation is highly complex and we can't abroad it to the fullest. We based many of our assumptions on literature to put down the complexity and we understand that BSC have a positive effect (Belnap and Büdel, 2016) and at the same time a negative effect to the vegetation (Booth, 1941; Brotherson and Ruthforth, 1983; George et al., 2003; Johansen, 1993; Prasse and Bornkamm, 2000; Zhang et al., 2016, 2010). At the same time, vegetation provides a positive effect to the BSC (Bowker, 2007), and because photosynthetic organisms compete to each other for resources, a negative effect is also expected. At last, the biopedturbations made by fossorial birds work as ecosystem engineers (Jones et al., 1994), and at a landscape scale the presence of ecosystem engineer would result in an increase of the species richness, along with the Competitive exclusion principle of Gause (Palmer, 1994) coexistence is allowed, and as a result vegetation increases its abundance and richness in an indirect way.

## 5 Conclusion

We can summarize the net of relationships in a high order interaction, where an interaction between two species is modulated by a one or more species (Bairey et al., 2016). In the present work we describe the relation between assemblages instead of only species. Fossorial birds through their biopedturbations attenuate the negative effect of BSC to the plant community, modifying the effect toward the vegetation without having a direct effect on it. This high order interaction plays an important role in determining diversity and allowing the coexistence of communities with multiple species. The impact of natural disturbances on BSC becomes more important in the transition between the dry season to the wet season and vice versa, the extreme seasonality would be softened by the biopedturbations. When moisture in soil cover with BSC isn´t enough for seeds to germinate and establish, the moisture in the biopedturbations would help the vegetation prevail longer. This high order interaction increases the special and temporal heterogeneity, important to the structure and functionality of the ecosystem, especially in the diversity maintenance (Huston, 1994; Pickett and White, 1985).

Data availability. The data is available in the supplement of this article

**Author contribution.** MR and CA designed the research. MR collected samples from the field and did all the measurements. MR and CA analysed the data. MR prepared the manuscript with corrections from CA.

**Competing interests**. The authors declare that they have no conflict of interest.

**Special issue statement**. This article is part of the special issue "Biological soil crust and their role in biogeochemical processes and the cycling". It is not associated with a conference.

**Acknowledges.** We are grateful to Mery Suni, Tomás A. Carlo and friends of the Ecology Department of the Natural History Museum for the invaluable help and support in the development of this work. We also thank the Natural History Museum of Lima and the heads of the National Reserve of Lachay for the institutional support. Furthermore, the present work was part of the undergraduate thesis of Maria C. Rengifo, and was possible thanks to the financial support of the Universidad




Nacional Mayor de San Marcos through the Vicerectorado de Investigación and the Instituto de Ciencias Biológicas Antonio Raimondi.

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





**Tables**

**Table 1: Chemical characteristics of the soil treatments. Different letters show statistical significance.**

| Treatment | Removed soil | | | Underlying soil | | | BSC | | | *P*-value |
|---|---|---|---|---|---|---|---|---|---|---|
| | Mean | SEM | | Mean | SEM | | Mean | SEM | | |
| EC dS/m (1:1) | 1.99[a] | ± | 0.50 | 0.67[b] | ± | 0.34 | 0.40[b] | ± | 0.08 | 0.043* |
| CaCO$_3$ % | 4.67[a] | ± | 0.67 | 2.02[b] | ± | 0.39 | 2.45[b] | ± | 0.31 | 0.043* |
| pH (1:1) | 8.53 | ± | 0.15 | 8.53 | ± | 0.44 | 8.27 | ± | 0.08 | 0.165 |
| OM% | 0.27[a] | ± | 0.13 | 0.38[a] | ± | 0.05 | 0.67[b] | ± | 0.08 | 0.043* |
| K (ppm) | 242.00 | ± | 6.00 | 379.33 | ± | 13.61 | 428.67 | ± | 99.93 | 0.165 |
| P (ppm) | 30.30 | ± | 6.91 | 22.53 | ± | 2.94 | 30.77 | ± | 2.17 | 0.165 |

(*) Significant values, p<0.05. Median test for two independent medians.



**Table 2: Plant density (ind/ m² ) of soil treatments in seedbank germination experiment and seedlings emergence observations.**

| Species | Seedbank germination experiment | | | Seedlings emergence observations | | |
|---|---|---|---|---|---|---|
| | BSC | Active disturbances | Inactive disturbances | BSC | Active disturbances | Inactive disturbances |
| *Cistanthe paniculata* | 2365.6 | 603.7 | 1564.7 | 60.1 | 118.8 | 50.0 |
| *Fuertesimalva peruviana* | 188.5 | 318.5 | 155.9 | 22.6 | 88.2 | 72.2 |
| *Chenopodium petiolare* | 231.1 | 18.5 | 70.6 | - | - | - |
| *Cryptantha limensis* | 82.0 | 14.8 | 61.8 | 0.3 | 2.1 | 6.3 |
| *Calandrinia alba* | 45.9 | 3.7 | 91.2 | 0.0 | 0.7 | 2.1 |
| *Nolana humifusa* | 57.4 | 18.5 | 38.2 | 0.0 | 2.1 | 4.2 |
| *Solanum phyllanthum* | 1.6 | 3.7 | 79.4 | 0.0 | 0.7 | 3.5 |
| *Oxalis lomana* | 36.1 | 14.8 | 5.9 | - | - | - |
| *Exodeconus prostratus* | 3.3 | 14.8 | 11.8 | 0.0 | 0.0 | 0.7 |
| *Palaua rhombifolia* | 13.1 | 0.0 | 14.7 | 0.0 | 0.7 | 0.7 |
| *Fuertesimalva limensis* | 18.0 | 3.7 | 0.0 | - | - | - |
| *Stenomesson coccineum* | 3.3 | 0.0 | 2.9 | - | . | . |
| *Solanum multifidum* | 1.6 | 0.0 | 0.0 | - | - | - |



**Figures**

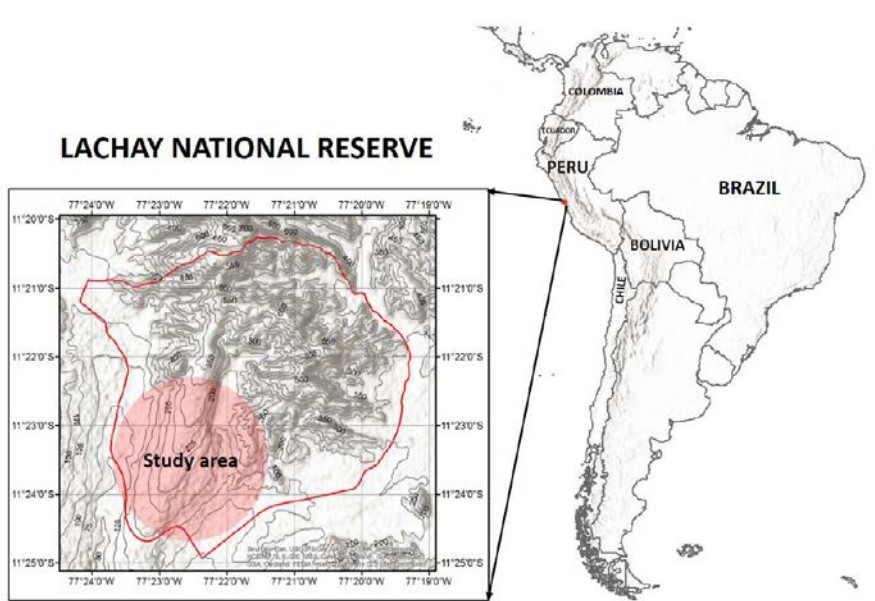

Figure 1: Study area in the Coastal desert of Peru.



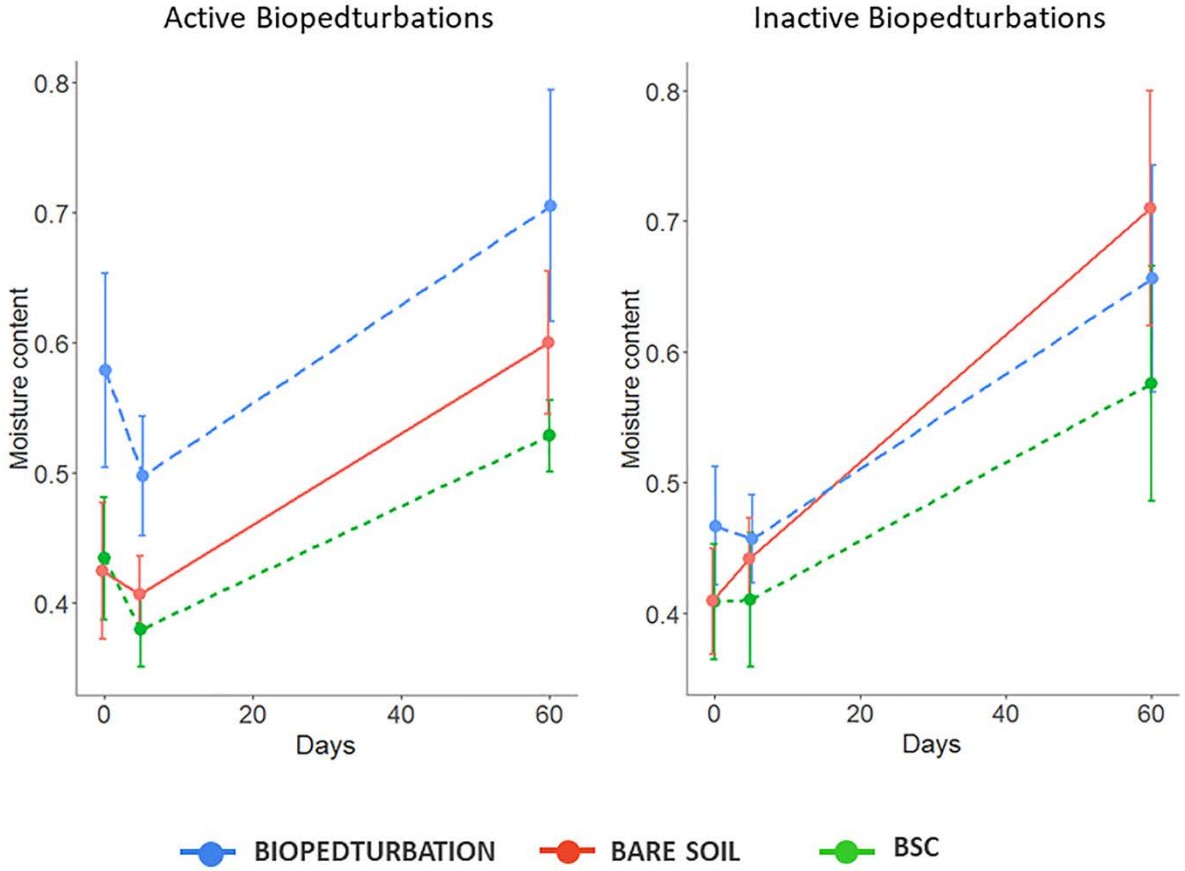

**Figure 2: Soil moisture content (%) of the two types of disturbances (Mean ± standard deviation). The comparison between the three treatments (Disturbance, bare soil and BSC) reflects the interaction between treatments and time (General Linear Model, F = 13.360, p = 0.000). At day 0 the bare soil samples were taken with the undisturbed biocrust layer, and immediately after the collection the biocrust was removed. Moisture content in active biopedturbations was higher than in inactive biopedturbations (Median test, p = 0.021).**



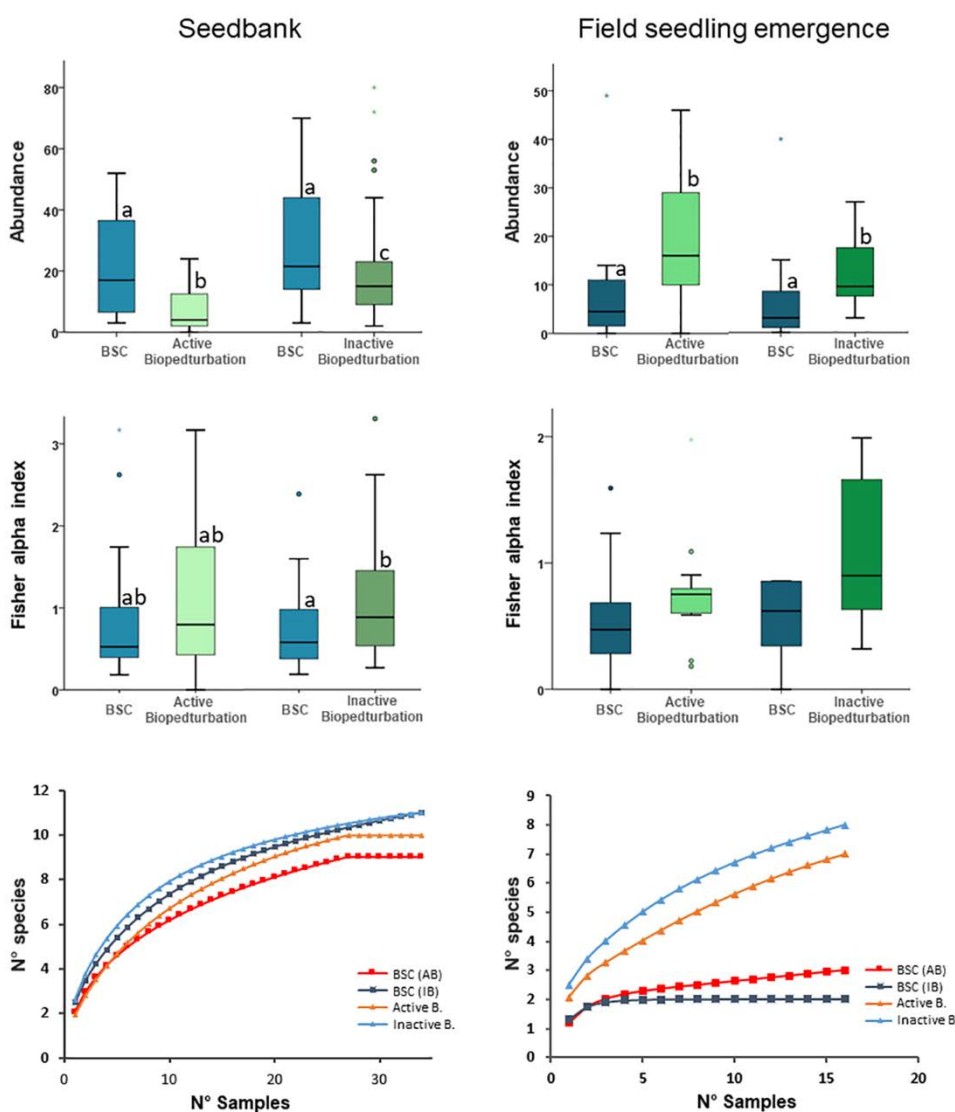

**Figure 3: Abundance, diversity Fisher alpha index and rarefaction curve of paired treatments of disturbances and biocrust. Different superscripts indicate a significant difference between BSC and active or inactive disturbance at P < 0.05. AB: Active Biopedturbation. IB: Inactive Biopedturbation. In the seedbank experiment BSC is more abundant than active biopedturbations (Wilcoxon test, p = 0.001), and inactive biopedturbations (Wilcoxon test, p = 0.046). Fisher alpha diversity index was higher only in inactive biopedturbations (Wilcoxon test, p = 0.035). In field emergence observations, BSC was less abundant than active biopedturbations (Wilcoxon test, p = 0.011) and inactive biopedturbations (Wilcoxon test, p = 0.020).**