# Peer review of "Disturbances of Biological Soil Crust by fossorial birds increase plant diversity in a Peruvian desert"

_Biogeosciences, 2017_

## Referee Comment (RC1) · D. Eldridge (Referee) · 16 Oct 2017

This manuscript describes a study of the impacts of disturbance of biological soil crusts on soil chemical properties and soil seed banks in a hyper-arid environment in Peru. This is an interesting topic, that has rarely been discussed in the literature. The research outlined in this paper is certainly worthy of publication in Biosciences, but in my opinion, the manuscript in its current state is not ready for peer review. It needs more conceptualisation in the introduction and more thought regarding the structure and experimental design, as well as a dedicated statistical section before I could assess the veracity of the results.

[Figure]

Firstly, the introduction is not very well developed and should comprise about four paragraphs. The first should identify the broader framework or a conceptual question or problem within this literature, then transition in a general sense into how your study might provide some answers. The second should define and introduce only the most important features of the study system or organism. In the third paragraph you should introduce your study system and explain the key features and why it is ideal for studying this question. The fourth should outline broad hypothesis and a handful of more specific hypotheses or questions, why the work is novel or important.

The methods section need substantially more work and are quite confusing. You need to say somewhere that you had three surfaces. 1: undisturbed biocrust; 2: biocrust disturbed by birds (hereafter the disturbed); 3: artificially disturbed plots (hereafter human disturbed) and then what these disturbances look the disturbed area of mound or a depression?. It is unclear how the 26 plots are distributed among the four treatments. Does one site consist of the four treatments? This setup is very confusing to the reader needs more explanation and perhaps a figure. Also, it is unclear what happens at day 60. How do you sample undisturbed crust at day 60. If you have already removed it?

It seems that if you are using mounds as your measure of soil disturbance, then you are essentially measuring subsoil. So if the bird digs beneath a piece of intact. Biocrust, then it should be exactly the same as your existing Biocrust, except that it will be older and have a greater chance of being (mound) or gaining resources (pit). Overall, the sampling units need more in-depth discussion in the introduction to describe what you expect in nature relative to different units and why you expected it. Otherwise, it's very hard for the reader to see where the manuscript is heading.

Section 2.2: why paired both active and in active with two paired control samples? Why not just look at three treatments (active, inactive and biocrust) and compare them with an over all, any modelling to look for significant differences? What exactly is an in active disturbance?
I would be inclined to describe the activity or process of animal disturbance as bioped-turbation then refer to the structures as disturbances.

---

## Referee Comment (RC2) · G.ÂăJ. Kidron (Referee) · 1 Nov 2017

The MS describes the effect of biopedoturbation on species diversity and plant germination in the Peruvian Desert. By disturbing the surface, fossorial birds create micro habitats that affect plant diversity and density. The increased diversity with increased heterogeneity is rather expected and the current MS adds additional information to a relatively large bulk of literature that exists on this topic. Nevertheless data from the Peruvian Desert is an important contribution. However, unfortunately, the MS is premature. While the authors describe the changes in seed bank and species diversity, no satisfactory explanation for this phenomenon is provided. This is a major obstacle once

publication in a leading journal is sought. The MS suffers from additional drawbacks (lack of data, unclear Methodology), while the structure, flow and choice of citations also need major improvements.

Main points 1. The topic presented is not new and adds to many other publications on biopedoturbation, as thoroughly summarized by Whitford and Kay (1999). Whereas the authors try to link the findings to the presence of biological soil crusts (BSCs), the data presented are fragmentary and not convincing. The authors suggest that higher moisture availability at the mounds and lack of runoff there may explain the higher diversity and biomass at the mounds. Yet, the authors (1) present only three dates with moisture data throughout the entire growing period (2) the authors assume a linear increase in moisture from day 5 to day 60 (Fig. 2) although intensive fluctuation in moisture is expected due to the erratic nature of precipitation in deserts (3) no rain data are provided, which does not allow for a proper evaluation of the data (4) the values used for the moisture are not clear (gravimetric? volumetric? ratio of WHC?). 2. The data should include a detailed account of the research site (general description of the geomorphology and/or dunes; the particle size distribution, i.e., the amount of sand, silt and clay; the main microorganisms within the BSC and possible their chlorophyll content in mg/m2 as well as plant cover) and description of the disturbance (are the three birds mentioned have the same disturbance? What is the size of the mounds?). It should be accompanied by photographs that show the research site with the BSCs, the mounds created by the birds and photographs from the experimental design. Long-term precipitation at the site, including the possible contribution of fog and dew (approximate precipitation) should be added. In addition, clear hypotheses should be outlined and the rationale for measuring each of the variables should be thoroughly explained (for instance, what is the rationale of the chemical analysis of the crust? of measuring the calcium carbonate or EC?). The Methodology should be thoroughly explained (statistics included). For instance, how do the authors define and differentiate between active and inactive mounds? Also, the methods or devices used for measuring each variable should be indicated, as well as the nutrient species. For instance: did the authors examine total or available P? 3. The Ms structure. Generally, the flow should be substantially improved. In essence the MS lacks Introduction. The Introduction should include general theories regarding the effects of disturbances on the ecosystem, with a specific emphasis on deserts and BSCs, especially on sand-covered BSCs. The Discussion should focus on the findings, discussing the similarities/differences with previous publications and the implications for the ecosystem. For instance, it is generally assumed that water availability at the mounds is lower (Moorhead et al., 1985), in contrast to the authors' conclusion. This should be thoroughly discussed. Also, the analysis is not clear. For instance, there are two main variables that may negatively affect moisture: loss of water due to runoff or increased evaporation. Both possibilities should be discussed. 4. The choice of references is unclear. The link between the mentioned topic and the references should be improved. Reports and abstracts should be avoided unless no other material exists. Book chapters and review papers should at best accompany peer review journal articles with empirical data (rather than being used as central references). Grouping together many topics (6th and 7th line in the Introduction) cannot guide the reader. Citations that refer to trivial points should be eliminated. 5. Many of statements do not reflect the state-of-the-art knowledge and the picture that emerges is rather simplistic. For instance, do crusts necessarily promote plant survival (section 1 of Introduction)? Are BSCs necessarily hydrophobic (upper p. 6)? Will buried crust 'stop' infiltration (upper p. 6)? Do BSCs loose their water following the consumption of water by the microorganisms?

---

## Referee Comment (RC3) · S. Chamizo (Referee) · 1 Nov 2017

This manuscript analyses the effect of biopedturbations on soil properties and plant abundance and diversity compared to BSCs, which is a very uncommon topic in the scientific literature regarding BSC and which makes the content of the manuscript interesting and novel. However, I have some concerns about the design of the experiment and discussion of the results obtained, especially regarding the effect of BSC on plant diversity and abundance. I have three main points to highlight: First, different plots have been set up and samples have been collected from different places for soil moisture, chemical properties and seedling analyses, so it makes difficult to establish direct

relationships between soil properties and abundance and diversity of plants in BSCs and biopedturbations. Nevertheless, if results about soil properties and seed bank and seedlings are presented, authors should make an effort to discuss these results in an integrated way, trying to link, to some extent, the effects on soil moisture and chemical properties in BSCs and biopedturbations with the results obtained regarding plant abundance and diversity. As written now, the Discussion looks like different paragraphs addressing independent results and without linking one result with others. Second, for the chemical properties, seed bank and seedling emergence experiments (it is not clear to me if also for the soil moisture), it has been compared BSC and the removed soil by biopedturbation (which is on top of a BSC), but not undisturbed soil devoid of BSC (or bare soil). This is important to really understand the effect of BSC on soil properties and plant establishment, as compared to bare soils. Third, it is said that BSCs have a negative effect on plants, but this conclusion is not clear to me from the experiments conducted and the results obtained as, on one hand, there is no comparison of the BSC with bare soil to clearly understand the effect of the BSC and, on the other hand, the disturbed soil (by biopedturbation) lies on the BSC and thus, the BSC might have indirect effects on seedling emergence by contributing with longer moisture retention and higher nutrient release to the mound of sand. The authors should discuss these points adequately in the manuscript. As a general comment, the language of the manuscript should be thoroughly revised by an English native speaker. More detailed comments are: Page 4, MM. Were samples for soil moisture, chemical properties and seedlings determination taken in areas next to each other? Also, indicate the period in which soil sampling was done (dry or wet season). Page 4, P10: What does "experimental plot" (after "the bare soil plot") mean? Please, explain the meaning of "active" and "inactive" biopedturbations. Page 4, P15. It is said that soil moisture was measured at three times, but what do days 0, 5 and 60 represent? If they are three independent measurements at three different times, it should be represented as time 1, time 2 and time 3 (or by the date) but not as a cumulative time since an initial time. In addition, the sentence "At day 0 the bare soil plot was sample with the undisturbed

BSC layer, and immediately after the collection of the sample the BSC was removed" is not understandable. Was the bare soil plot soil devoid of BSC or soil with BSC in which the crust was removed? In the latter case, it is not measured soil moisture content in bare soil but in the soil underneath the crust. The soil below the BSC usually has better properties (higher EPS, N, aggregation. . .) than the bare soil and thus, soil moisture is likely higher in the soil beneath the crust than in adjacent soils devoid of BSC. If the BSC was removed from the soil and water content was measured in the underlying soil in day 0, what was measured after 5 and 60 days? Soil moisture in the scalped soil? Both measurements are not comparable because in day 0 the presence of the BSC conditions soil moisture in the underlying soil, while in the resting days, soil water content is measured in soil lacking the BSC. I think authors should show soil moisture data only for the days in which similar surface types are compared, and in the case of the bare soil, let clear that it consists of scalped soil where the top BSC was removed (which is not the same than bare soil).

Regarding the method, if moisture content is determined by weight, it is gravimetric water content (g H2O/kg soil), not volumetric water content. Page 4, P20. Which soil depth was sampled for the analysis of chemical properties? Besides, the method used for the determination of each soil property should be explained. Page 4, P25. Please, homogenize the terminology for biopedturbation samples as different terms are used along the text ("biopedturbation plot", "removed soil from the loose soil of the entrance of the burrows", "disturbed soil"). Page 5. It should be included a section of "data treat-ment" or "statistical analyses" to explain how statistical differences were analysed and also to explain the indices of plant abundance or diversity used. Page 5. In general, description of the results is very poor and should be greatly improved. Authors should describe more in detail differences in the properties analysed between BSC and pi-opedturbations. Page 5, P5. Please, describe first differences in soil moisture among times (also explains what the different times represent), and then, differences among "bare soil" (see my comment above), BSC and types of biopedturbations. Page 5, P10. Results of chemical properties should be better described by comparing the BSC with

the underlying soil and both with the disturbed soil for all soil properties. For instance, it could be shown average values of the different properties in BSCs compared to disturbed soil, Pages 5-6. The Discussion should be substantially improved. Authors should make an effort to connect the different results obtained and, for instance, try to link the results of seed bank and seedlings with the results of soil moisture, organic matter and nutrients in BSC and biopedturbations. The manuscript should be also improved by comparing with other published studies that analyse the effect of biopertudbations on seedlings and by adequately explaining and discussing the positive and negative effects of BSC on seedling and plant establishment, and relating these effects with their effects on soil properties. I also recommend using more recent references in the Discussion as some of them are old and there is a large budget of articles recently published about the influence of BSC on soil properties (water content, nutrients...). Page 6, P5-10. In this paragraph it is said that in the piopedturbation, the mound of sand is on top of the BSC. If the BSC acts as a seal on the soil surface limiting water infiltration into deeper soil, it could have a positive effect retaining moisture at the surface and keeping moisture longer in the sand above it, indirectly favouring seedling in the mound of sand. Page 6. Soil moisture, chemical properties and seedlings in biopedturbations and BSC are discussed separately, and no relationships and interactions between these properties have been discussed. For instance, higher organic matter and nutrients in BSC could be the reason for higher plant abundance. In contrast, lower moisture could be the reason for lower diversity, as only certain species better adapted to drier conditions could be competitive for growing in soils covered by BSCs, while others with more water requirements would grow better in biopedturbation-disturbed soils. Page 6, P25-40- An important point to discuss is the different results found in the seedling greenhouse experiment and the field experiment. Such differences could be related to differences in water availability between both experiments that could strongly condition species diversity and abundance in the BSC under greenhouse and field conditions. In the greenhouse, samples were irrigated frequently and in this case, higher abundance of the seed bank was found in the BSC compared to biopedturbations,

while in the field, with limited water availability, opposite results were found. Water, thus, appears to be a major driver for seedling abundance. This should be discussed in the Discussion. Page 6, P30-35. Together with moisture availability, I really think that the reduction in seedling emergence in BSC is greatly associated to a physical impediment: the seal created by the crust impedes seed penetration and leaves the seed more exposed (and less protected) to hostile environmental factors, at the same time that facilitates seed removal by wind. Page 7, P5-10. This paragraph is confusing and mix different ideas about BSC and plant interactions. The authors should explain along the Discussion the contrasting effects of BSC on vegetation, and why they can have positive and negative effects on vegetation. The sentence "At the same time, vegetation provides a positive effect to the BSC (Bowker, 2007), and because photosynthetic organisms compete to each other for resources, a negative effect is also expected" is not understandable and contradictory as it suggests a simultaneous positive and negative effect of vegetation on BSC. I do not think plants and BSC compete for water and nutrient resources, but that BSCs grow in the areas where water and nutrients are not available enough to allow plant establishment.

The sentence "and at a landscape scale the presence of ecosystem engineer would result in an increase of the species richness, along with the Competitive exclusion principle of Gause (Palmer, 1994) coexistence is allowed, and as a result vegetation increases its abundance and richness in an indirect way." is very abstract and not understandable in this context. Please, either rewrite it or delete this sentence.

Page 7, P20. What do the authors mean by "relationships of a high order interaction"? It is not clear that BSCs have a negative effect on the plant community and that "biopedturbations attenuate the negative effect of BSC to the plant community". Likely, BSCs could have an indirect effect on the disturbed soil by maintaining soil moisture longer and by contributing nutrients to the mound of sands. In addition, it has not been analysed seedling abundance and diversity in BSCs compared to bare soil. Some editing comments: Is the term "biopedturbation" more commonly used than "bioturbation"?

The second one is more familiar to me. Page 4, P35. "We consider 15 replicates for each type of biopedturbation" Page 5, P15. "...where the BSC samples had a higher abundance of germinated seeds than active and inactive biopedturbations." Figure 2. Include the units for soil moisture in axis Y. In addition, units in the legend seem to be wrong (gH2O/kg soil, not

---

## Author Comment (AC1) · 19 Dec 2017

We thank gratefully the comments given, which we found constructive and improved tremendously the quality of this manuscript. We agree with most of the comments. We have revised the manuscript in the light of the comments. Below the separated specific comments we indicate our responses and we attached a new version of the manuscript in the supplement.

Comments by D. Eldrigge

This manuscript describes a study of the impacts of disturbance of biological soil crusts

on soil chemical properties and soil seed banks in a hyper-arid environment in Peru. This is an interesting topic, that has rarely been discussed in the literature. The research outlined in this paper is certainly worthy of publication in Biosciences, but in my opinion, the manuscript in its current state is not ready for peer review. It needs more conceptualisation in the introduction and more thought regarding the structure and experimental design, as well as a dedicated statistical section before I could assess the veracity of the results.

Specific comments

Firstly, the introduction is not very well developed and should comprise about four paragraphs. The first should identify the broader framework or a conceptual question or problem within this literature, then transition in a general sense into how your study might provide some answers. The second should define and introduce only the most important features of the study system or organism. In the third paragraph you should introduce your study system and explain the key features and why it is ideal for studying this question. The fourth should outline broad hypothesis and a handful of more specific hypotheses or questions, why the work is novel or important.

RE: We restructure the Introduction to meet all the points given.

The methods section need substantially more work and are quite confusing. You need to say somewhere that you had three surfaces. 1: undisturbed biocrust; 2: biocrust disturbed by birds (hereafter the disturbed); 3: artificially disturbed plots (hereafter human disturbed) and then what these disturbances look the disturbed area of mound or a depression?

RE: We add a section to described the biopedturbations in the Introduction and stated the different surfaces analyzed in each method section

It is unclear how the 26 plots are distributed among the four treatments. Does one site consist of the four treatments? This setup is very confusing to the reader needs more

explanation and perhaps a figure. Also, it is unclear what happens at day 60. How do you sample undisturbed crust at day 60. If you have already removed it?

RE: We add a paragraph in the Introduction as a first description of the biopedturbations studied and further description in the third paragraph of the 2.1 Method section. To help visualize the description of the experimental design of the moisture sampling we add a picture (Fig 3). To simplify the data analysis and meet our objectives, we also shortened the soil moisture content analysis, and only used the data taken at day 60. Also, we were able to sample undisturbed crust three times in the same plot because the 100 gr soil sample only represent a small area and volume of the total 30x30 plot, and at the second and third time we sampled in the sample plot but not in the exact same point in the plot.

It seems that if you are using mounds as your measure of soil disturbance, then you are essentially measuring subsoil. So if the bird digs beneath a piece of intact. Biocrust, then it should be exactly the same as your existing Biocrust, except that it will be older and have a greater chance of being (mound) or gaining resources (pit). Overall, the sampling units need more in-depth discussion in the introduction to describe what you expect in nature relative to different units and why you expected it. Otherwise, it's very hard for the reader to see where the manuscript is heading.

RE: Done. We add a better description of the biopedturbations in the introduction, as well as addressing it better trough the discussion.

Section 2.2: why paired both active and in active with two paired control samples? Why not just look at three treatments (active, inactive and biocrust) and compare them with an over all, any modelling to look for significant differences? What exactly is an inactive disturbance? I would be inclined to describe the activity or process of animal disturbance as biopedturbation then refer to the structures as disturbances

RE: In order to diminish the environmental heterogeneity given by series of factors that we cannot control, we used paired both active and inactive biopedturbations with

their own control samples, since its very unlikely to find an active bioped really close to an inactive biopedturbation and be able to replicate this. At the same time, the nonparametric paired analysis is a more strong a robust statistical tool considering the design and the low sample size.

María Cristina Rengifo

Please also note the supplement to this comment:
https://www.biogeosciences-discuss.net/bg-2017-376/bg-2017-376-AC1-supplement.pdf

**Supplement:**

**Disturbances of Biological Soil Crust by fossorial birds increase plant diversity in a Peruvian desert**

María Cristina Rengifo[1,3], Cesar Arana[1,2]

[1]Departamento de Ecología, Museo de Historia Natural de la Universidad Nacional Mayor de San Marcos, Lima, 15072, Perú.
[2]Laboratorio de Ecología y Biogeografía Terrestre de la Facultad de Ciencias Biológicas de la UNMSM, Lima, 15081, Perú.
[3]School of Forestry, Northern Arizona University, 200 E. Pine Knoll Drive, Flagstaff, AZ 86011, USA.

*Correspondence to*: Maria C. Rengifo (mcr335@nau.edu)

**Abstract.** The Lomas Formation are fog-dependent oases within the hyper arid band of the Peruvian coast. Biological soil crusts (BSC) form in the Lomas and interact with their fauna and flora. Here we asked if natural disturbances – biopedturbations - made by fossorial birds have an effect on seedling emergence in the Lomas Formations in the National Reserve of Lachay in Lima, Peru. We analysed active and inactive avian biopedturbations, undisturbed BSC and scalped soil field samples for moisture content, soil chemical properties and the seedbank and the field emergence of seedlings. Active biopedturbations had the highest soil moisture content and BSC showed the lowest values. Organic matter content was significantly higher in the BSC than the soil beneath it and the bare soil. However, $CaCO_3$ content and EC were higher in bare soil than the other treatments, and no significant differences were found in soil pH, phosphorus or potassium content between treatments. In the seedbank experiment, 13 herbaceous plant species were found; furthermore, biopedturbations had a higher diversity but lower abundance than the BSC. However, in the field observations biopedturbations had a higher diversity and abundance of seedlings than BSC and only 8 herbaceous species were found. The species *Fuertesimalva peruviana* (L.) Fryxell, *Exodeconus prostratus* (L'Hér.) Raf., *Cryptantha granulosa* I.M. Johnst. *Solanum phyllanthum* Cav. and *Calandrinia alba* (Ruiz & Pav.) D.C increased their abundance in biopedturbations. Our results showed the positive effects on seed germination and diversity of vascular plants by the natural disturbances made by fossorial birds in a unique ecosystem of the Peruvian desert, and demonstrates the importance of spatial and temporal heterogeneity for ecosystem structure and functioning.

**1 Introduction**

Natural disturbances and their impact in ecological processes has been broadly studied in drylands, and focused extensively on small mammals (Eldridge et al., 2012; Hobbs and Huenneke, 1992; Kerley et al., 2004; Schooley et al., 2000; Whitford and Kay, 1999). Soil disturbances made by burrowing animals directly modify habitats and modulate the availability of resources;
5   by which they are consider ecosystem engineers (Guo, 1996; Hansell, 1993; Jones et al., 1994; Wright et al., 2004). Although it's known that burrowing mammals contribute to the heterogeneity that support different plant communities at small and large scales in landscapes (Eldridge et al., 2012; Whitford and Kay, 1999), very little is described on fossorials birds with similar behavior on drylands.

In the hyperarid system of the Sechura-Atacama Desert, there is a fog oasis known as *Lomas* (Fig 1; Rundel et al., 1991b).
10   Despite the extreme climatic conditions, the coastal hills in *Lomas* are high in biodiversity and endemism of plants and animals (Dillon and Rundel, 1990; Ferreyra, 1983; Pulido et al., 2007). The dry conditions limit the establishment of perennial plants and allow the development of annual vegetation only in the winter months when marine fog and fine drizzle provide water (Ferreyra, 1953). Common birds found in the central Peruvian *Lomas* generate soil disturbances –biopedturbations-. Miners *Geositta* spp. and the burrowing owl *Athene cunicularia* excavate tunnels and create notable mounds in the landscape. These
15   disturbances occur mostly in the low elevation areas of the coastal hills (between 150 and 400 meter) where vegetation cover is limited and the soil cover is dominated by biological soil crust (i.e., BSC) (Ferreyra, 1953). BSC in the central Peruvian *Lomas* has been recently reported for the first time (Arana et al., 2016). A knowledge gap regarding the basics of biocrust ecology, specifically basic structural and taxonomic characterization, nutrient fluxes and interactions with higher taxa, needs to be filled.

20   This soil community is known as important components of arid ecosystems (Johansen, 1993; Li, 2012; West, 1990). Biological soil crusts found around the world increase the fertility of soils and affect the physical properties of it by altering water infiltration, runoff, albedo, and temperature (Belnap et al., 2016; Bowker et al., 2010; Prasse and Bornkamm, 2000; Weber et al., 2016; West, 1990; Zaady and Shachak, 1994). As a result of those effects, the emergence and survival of vascular plants can be promoted (Jones et al., 1994, 1997).  Although effects on vascular plants are more variable and have negative effects
25   are also found depending on BSC and plant characteristics (Boeken and Shachak, 1994; Bowker, 2007; Li et al., 2010), which develops a greater complex scenario for biocrust and vascular plants interactions.

In order to create a basic understanding of the interactions between the components of the *Lomas* ecosystem, we look into the hyperarid system dominated by biocrust and the plant community response to biopedturbations. We hypothesized that the biopedturbations will create a positive effect on plant diversity and might be linked to some abiotic factors. The soil removal
30   and generation of mounds would increase soil moisture content and nutrient availability and increase seed germination. We created a study to specifically test the effects of fossorial birds' disturbances on soil moisture content, physical characteristics, seedbank germination and the emergence of annual plants. The study provides new information on the complexity and functioning of this understudied ecosystem and quantifies the relevance of interaction among its components.

In our study we targeted biopedturbations generated by fossorial birds that are the major disturbing agent in our site. The
35   landscape seen in the lowest part of the hills (*Lomas*) has a flat topography with a narrow inclination that gets steeper going east as we get closer to the top of the hills. The vast area is fully covered with biological soil crust, and only during 4 to 5 months of the wet season we see the establishment of the annual vascular plants. The rest of the year the area lacks higher plants cover. The birds' biopedturbations create bare patches observed with the naked eye across the landscape (Fig. 2). When burrows were active, we targeted the disturbed mounds of soil that have a larger area than the burrow entrance. The
40   soil profile of the mounds has 3 basic layers: an underlying sandy soil, a biocrust layer and another soil layer on top from the

burrowing activities of the birds. When burrows are abandoned, the lack of bird activity allows the biocrust organisms in the area to colonize the disturbed top soil, and a new layer is added to the soil profile; and we consider this as the inactive biopedturbations.

**2 Methods**

2.1 Study site and biopedturbations

The Atacama Desert is a coastal desert that extends 3500 km, from the region north of Trujillo near the Ecuadorian border of Peru, to central Chile (Rundel, 1978). This desert owes its aridity to the persistent temperature inversion associated with the cool north flowing Humboldt Current and the generally stable position of the strong Pacific anticyclone. The Andes Mountains prevent moisture from the east (Houston and Hartley, 2003). The National Reserve of Lachay is located 105 km north from the city of Lima, in the central coast of Peru (S11°23.6', W77°23'). The reserve contains a unique fog and mist-fed ecosystem called Lomas Formations within the hyper arid band of the Peruvian coast (Fig. 1). The landscape is characterized by small hills that create a smooth gradient from 150 m to 750 m of altitude, a mean annual precipitation in the open of 168 mm yr$^{-1}$. and in the high humidity season, from July until September, a dense fog comes from the sea adds moisture to the hills allowing the establishment of endemic flora (Rundel et al., 1991a).

The study area is located in the lower part of the hills (S11° 23.87', W77° 23.13') in approximately 1.4 km$^2$ with a smooth gradient from 150 to 250 m of altitude with a mostly flat topography (Fig. 2A). The seasonal vegetation found in the humid season is characterized by the presence of herbaceous plants of rapid flowering. The sandy loam soil is covered by a dark biological soil crust dominated by cyanobacteria (Arana et al., 2016), which is a type of BSC commonly found in warm deserts (Belnap et al., 2001; Pietrasiak, 2014). The BSC is 1-5 mm thick (Fig. 2E), and also has a low percentage (~25%) of moss (*Bryum argenteum* Hedw mostly) and some crustose and fruticose lichens. In this landscape three species of fossorial birds are the main agents of biopedturbations: the burrowing owl (*Athene cunicularia*), the coastal miner (*Geositta peruviana*) and the greyish miner (*Geositta maritima*).

For the purpose of this study we targeted the mound of removed soil that is placed over the surface as the biopedturbation (Fig. 2A-C). The birds break the BSC to create their burrows and the removed soil is placed on top of biocrust. We aimed for the average mounds that had an area of approximately 0.6 m$^2$; mostly from miners and some small ones from the burrowing owls. In the study area mounds of the two species of miners are indistinguishable one from another and range from 0.08 to 0.7 m$^2$. The mounds of the burrow owl are usually bigger and range between 0.4 to 1.3 m$^2$ (Unpublished data). We targeted active and inactive biopedturbations, the active ones were considered from the active burrows where the activity of the birds keep the top soil of the mound loose. Inactive biopedturbations were the mounds that were colonized by an early successional biocrust as a result of an abandoned burrow.

2.2 Moisture content

We examined the top 5 cm of soil moisture content between active and inactive biopedturbations, biological soil crust and scalped soil (Fig. 3). We established 26 experimental sites, 15 for active biopedturbations and 11 for inactive biopedturbations. Each experimental site consisted of three 30 x 30 cm plots placed not more than 1 m of separation: a scalped soil plot, a biopedturbation plot, and a BSC plot. The scalped soil plot was an 30x30 cm area where the surface layer

of BSC was removed to simulate bare soil surface. The biopedturbation plot was an area marked on the mound of sand, and the BSC plot was an area with undisturbed BSC. Selected biopedturbations were separated at least 10 m.

After two months of the BSC removal in the scalped soil plot, we sampled 100 g of the top 5 cm of soil inside each plot. Samples were collected in hermetic bags, and later in the laboratory were weighted, dried in an oven for 24 h at 105 °C, reweighed and their gravimetric moisture content calculated (Yair et al., 2011). Each sample represented a small fraction of the 0.09 m$^2$ area. The sampling collection was made at the end of October of 2015. This was an anomalous year influenced by the ENSO, 0 mm of precipitation was register in October, when 0.2 mm was expected.

$$Water\ content\ (\%) = \frac{w_w - w_d}{w_d} x\ 100$$

Where $W_w$ is the weight of wet soil in grams and $W_d$ is the weight of dry soil.

**2.3 Soil chemical properties**

To examine the soil chemical properties of the different layers of a biopedturbations we considered three soil treatments: (1) The undisturbed layer of BSC, (2) the underlying soil of 5 cm deep which was directly below the initially undisturbed layer of BSC sampled, and (3) the biopedturbation soil from the loose sand of the mound. Each treatment was replicated three times. The biopedturbations were separated by at least 10 m, and we used the undisturbed BSC next to each biopedturbation to diminish environmental heterogeneity. Every soil sample weighted approximately 500 g. The routine soil analysis included the available phosphorus (P), exchangeable potassium (K), calcium carbonate (CaCO$_3$), soil organic matter (O.M.), pH and electrical conductivity (E.C.), and was made by the Water, Soil and Environment Analysis Laboratory (LAASMA) of the Universidad Nacional Agraria La Molina, in Lima.

**2.4 Seedbank evaluation**

To evaluate the seed bank between biopedturbations and the soil covered with BSC, we took paired samples of soil from the mounds and from the undisturbed BSC next to the mound. Each soil sample was taken from a 10x10 cm area and 5cm deep; the paired samples were separated by approximately 1 m. We used 27 active biopedturbations and 34 inactive biopedturbations, each paired with an undisturbed BSC sample.

We conducted the experiment in a greenhouse at the UNMSM. The bagged soil samples were placed in plastic trays and arranged in a Latin Square design to eliminate positional effects within the greenhouse. The trays were watered regularly every two days with tap water, and the number of seedlings were recorded at frequent intervals. The germination beyond five weeks was found negligible and during the germination period there was no soil disturbances and seedling were removed only at the end of the experiment. The taxonomic determinations of species were made with dichotomous keys and specialized bibliography (Fryxell, 1996; Krapovickas, 2007; Lleellish et al., 2015; Sagástegui and Leiva, 1993; Tate, 2011).

**2.5 Field seedlings emergence**

We established a new set of plots in the field to observe the natural seedling emergence. We marked 30 x 30 cm paired plots, one on the biopedturbation and the other one in the undisturbed BSC, separated 1 m from each other. Paired sampling was need to diminish the environmental heterogeneity and compare plots with similar conditions due their proximity. We consider 15 replicates for active biopedturbations and 15 for inactive ones. In every plot we count and identify all the present seedlings. We evaluated the seedlings emergence at the beginning of the wet season in August 2016.

**2.6 Statistical analysis**

To determine if mounds, bare soil or BSC had different soil moisture, we used nonparametric related-samples Wilcoxon signed-rank test, to compare the median of paired samples. Soil chemical properties were compared in pairs with the nonparametric Independent-sample median test. Plant diversity was calculated with Fisher's alpha index defined by the formula S=a*ln(1+n/a) where S is number of taxa, n is number of individuals and a is the Fisher's alpha. We used PAST Paleontological Statistic software version 3.0 to extract the diversity index. Plant abundance values didn't go through any transformation. Plant diversity index and abundance values were analyzed by pairs using nonparametric related samples Wilcoxon signed-rank test. All the statistical nonparametric analyses were made in Software SPSS version 19, and α=0.05 for every case. Plant density was calculated by the sum of all seedling divide by number of samples and the area sampled in $m^2$.

**3 Results**

**3.1 Soil moisture content**

Soil moisture content showed different patterns among active and inactive biopedturbations. Active biopedturbations have the highest moisture content (0.71% ±0.164) among the soil treatments (Wilcoxon signed-rank test, p=0.041 paired to scalped soil, p=0.002 paired to BSC). On the other hand, inactive biopedturbations (0.66% ± 0.164) have similar moisture content than scalped soil (0.71% ± 0.152) and that BSC (0.58% ± 0.152). Biological soil crust showed to have the lowest moisture content in both types of biopedturbations (Wilcoxon signed-rank test, p<0.05 for active and inactive biopedturbations).

**3.2 Chemical properties**

Our results show that no significant difference in the potassium and phosphorous content among the BSC layer, the underlying soil or the biopedturbation soil (Independent-sample median test, p>0.05). The percentage of soil organic matter in the BSC layer (0.67%) was significantly higher than the underlying soil (0.38%) and the biopedturbation soil (0.27%), Independent-sample test p=0.043; and no difference was found between the organic matter values of the underlying soil and the biopedturbation soil. The pH values remained similar among the treatments. Calcium carbonate and electric conductivity values showed the same trend between treatments, where the highest values were found in the biopedturbation soil, and the BSC layer and the underlying soil had similar values (Table 1).

**3.3 Effects on plant germination**

We found 13 different native plant species germinated from the overall seedbank. Among treatments, active biopedturbations had a mean of 10.5 germinated seeds, significantly lower (Wilcoxon signed rank test, p =0.001) than the 27.4 germinated seeds in the paired BSC. Inactive biopedturbations had the same trend, with a mean of 21.2 germinated seeds, significantly lower (Wilcoxon signed-rank test, p=0.046) than the 33.1 germinated seeds of the paired BSC. (Fig 3). The Fisher's alpha diversity index of germinated species was 1.05, and significantly higher than the 0.76 of the paired BSC (Wilcoxon signed-rank test, p=0.035), but the same trend was not statistically supported for diversity in active biopedturbations. The number of species expected in the rarefaction curves was similar between treatments. Nevertheless, the BSC had a lower expected richness compared to their paired active biopedturbations (Fig. 5)

In field observations (Table 2) we recorded the emergence of only eight species at the beginning of the wet season. In contrast to the abundance found in the seedbank, the natural seedling emergence was higher in biopedturbations, active biopedturbations had a mean of 21.9 seedlings, significantly higher than the mean 11 seedlings in the paired BSC (Wilcoxon signed-rank test, p=0.011). Inactive biopedturbations showed the same trend, with a mean of 12.5 seedlings, significantly higher than the 8.2 seedling in BSC (Wilcoxon signed-rank test, p=0.020). The diversity of seedlings between treatments was not significantly different, although inactive biopedturbations had a slightly higher diversity index compared to the

paired BSC (Wilcoxon test, p=0.507). The number of expected species in the rarefaction curves are higher in both active and inactive biopedturbations compared to their paired BSC. Inactive biopedturbations shows the highest expected richness between treatments (Fig. 5)

The floristic composition (Table 2) shows that *Cistanthe paniculata* is the species with the highest density in every
5   treatment, except for the inactive biopedturbations in the field observations, where the species with the highest density is *Fuertesimalva peruviana*. We found species that have a higher density in inactive biopedtubations in both the seedbank and in the natural emergence: *Exodeconus prostratus, Solanum phyllanthum* and *Calandrinia alba*. An additional species, *Cryptantha limensis* also show a higher abundance in inactive biopedturbations in the field observations but without the same pattern in the seedbank.

10   **4 Discussion**

Our study is the first to look at how biopedturbation by fossorial birds alters the soil chemistry, moisture and potential for plant germination in the Lomas region of the Atacama Desert. This area is fully covered in dark cyanobacterial biocrusts except where burrowing activity has occurred. Burrowing fossorial birds are acting as ecosystem engineers, opening up niches for plant germination. Our work shows that biopedturbations had a positive effect on the plant community by
15   increasing the germinating plant diversity in the transition to wet season. Some annual plant species benefit from both active and inactive biopedturbations.

**Effects of biopedturbation on soil moisture**

Areas with BSC presented the lowest values of soil moisture content, that is a possible result of a low water infiltration, because the BSC tends to seal the soil surface (Brotherson and Ruthforth, 1983; Zhang et al., 2010). Factors like biological
20   soil crust characteristics, the topography and soil types (Chamizo et al., 2012) can cause BSC to increase (Brotherson and Ruthforth, 1983; Bu et al., 2015) or decrease (Eldridge et al., 2000; Gao et al., 2010; Kidron and Yair, 1997; Yair, 1990) infiltration rates in the soil. It was suggested that in sandy soils the BSC decrease the infiltration as a result from the reduction in the porosity (Warren, 2001). The biological soil crust of our study may decreased the water infiltration and as a consequence diminish the soil moisture content, this results also agreed with studies in the Negev desert (Eldridge et al.,
25   2000; Keck et al., 2016) which resembles physical characteristics of our studied area. Whereas the cyanobacteria and lichens of the BSC decrease the permeability of the soil (Loope and Gifford, 1972), the bare soil doesn't (Keck et al., 2016). Our data shows the same results, where the scalped soil had higher moisture content than BSC, that could be explained by the permeability, because biological soil crust tend to seal the soil surface (Booth, 1941; Brotherson and Ruthforth, 1983; George et al., 2003) and consume the water available in the most superficial soil layer (Bu et al., 2015; Gao et al., 2010)

30   On the other hand, biopedturbations present higher values of moisture content. In the case of active biopedturbations, moisture content was higher than the other soil surfaces, but the inactive biopedturbations show similar moisture content than the bare soil. We hypothesized that the soil profile of active biopedturbation (Fig. 3) allows water to infiltrate easily through the first layer of sand, and because of the hydrophobic characteristics and the water absorption of biocrust organisms (Kidron et al., 1999; Rodríguez-Caballero et al., 2013) in the buried BSC layer the moisture is better retained, compared to
35   the bare soil and the soil cover with BSC. Our results differed completely with a study in Northern Negev of Israel (Boeken and Shachak, 1994) that found that man-made mounds consistently presented the lowest moisture content compare to the BSC matrix, the main difference with this study is the characteristic of the man-made mounds, that were bigger and taller than the bird mounds. The soil moisture content on the first 15 cm was only loose soil, where in our study the first 5cm involves the BSC layer. The inactive biopedturbation, has an additional incipient BSC layer that could diminish the water
40   infiltration by reduction of porosity and the consumption of the water available in the most superficial soil layer and the (Bu et al., 2015; Gao et al., 2010; Warren, 2001), but not as much as a well develop BSC that shows the lowest values among treatments. As a result, the moisture content in inactive biopedturbations is similar to the scalped soil and to the BSC.

**Effects of biopedturbation on soil chemistry**

The soil chemical properties give us an approximation on the disturbance effect of the fossorial birds. Although many physical and chemical soil characteristics are insightful way to understand the ecological and biochemical processes occurring in the system, our study resources limited the extent of measurable characteristics. A basic routine soil analysis provided an insight of soil fertility and compared if the biopedturbations generate a great effect on those characteristics.

5    The nutrients potassium and phosphorus and the pH didn't show statistical differences between treatments, suggesting that the bird's biopedturbations do not alter significantly the nutrient soil content. Potassium values were slightly higher in the BSC, which is expected since it has been established before that BSC increases the soil fertility (Belnap and Harper, 1995; Harper and Pendleton, 1993). Though also bird droppings may contribute to soil fertility, our results did not support it. Differences in carbon or nitrogen content might be a better indicator of fertility, and thorough studies should be done in the
10   *Lomas* environment. Soil pH has been reported in other studies to not be significantly altered by the presence of BSC (Evans and Johansen, 1999; Guo et al., 2008; Kidron et al., 2015), and our results suggest the same. Soil organic matter was expected to be higher in the BSC in comparison with the underlying and biopedturbation soil, since is directly related to the organic carbon and BSC have high concentration of organism living in it (Delgado-Baquerizo et al., 2016; Guo et al., 2008). organic matter content, suggesting that neither the occasional bird droppings, nor the destruction and removal of the BSC
15   increases the organic matter in the other soil surfaces.

Unlike other studies (Guo et al., 2008), calcium carbonate content was not higher in the BSC, instead the biopedturbation soil had the higher values, this results may be explained by the abundant remains of gastropods shell's present in the locality, and not a reflection of increase nutrients in the soil. At the same time, since electric conductivity is influenced by the concentration of calcium carbonate BSC did not show high values of EC; despite being expected due to the concentration of
20   ions released by the decomposition of microorganisms. Although some differences in chemical properties where found and were expected, the lack of significant variation in K and P nutrients and pH between treatments shows that the biopedturbations might not have a greater effect on those soil characteristics due to the small area affected and may not have a great contribution in plant establishment.

**Biopedturbation effects on plant communities**

25   The plant community responded different in the greenhouse experiments than in the field emergence observations. Studies on BSC effects on vascular plants germination and establishment have contradictory results (Bowker, 2007) some consider that BSC can affect negatively plant density (Johansen, 1993; Prasse and Bornkamm, 2000), other showed the positive effect in semiarid ecosystems (Boeken et al., 2004; Defalco et al., 2001) and many species-specific effects (Hawkes, 2004; Su et al., 2009). By stop the limitation of water availability on the soil, the total germination of the seed bank in the greenhouse
30   represent the total viable seeds contained in the first 5cm of the soil (Thompson and Grime, 1979; Zhang et al., 2010). We found that biopedturbations are significantly less abundant in seeds of annual vascular plants in contrast with the BSC; nevertheless, are more diverse and rich in species composition. Even though a smooth biocrust increases the chances that seed may be taken by the wind (Belnap, 2006; Boeken and Shachak, 1994), plant litter annually accumulated on the soil surface remain along with a large seed load. The mounds of sand over the BSC do not present the same quantity of plant
35   litter and as a consequence it would be expected to have less seeds. A higher diversity in biopedturbations suggests a positive effect in the vegetation composition, but it would be necessary to study more thoroughly this phenomenon to understand if those biopedturbations work as a seed trap, if the higher diversity is product of the soil removal, or if the loose soil creates a better new topography compared to the tightly woven BSC surface (Boeken and Shachak, 1994).

Our field data taken in the transition from dry season to wet season shows that biopedturbations have higher abundance and
40   diversity in contrast with the BSC. Since the seedbank is more abundant in the BSC, the low emergence in the field compared to biopedturbations might be explained by the soil moisture content, that we found to be higher in the biopedturbations. And, although diversity was no significantly different, biopedturbations in the field show a higher species richness as seen in the seed bank.
Moreover, biopedturbations had a positive effect in some specific species: *Fuertesimalva peruviana,* in spite of been a high
45   density specie in the BSC, almost quadrupled its abundance in active and inactive biopedturbations in the field, and doubled

its seedbank density. *Exodeconus prostratus* the species with the lowest density in the field was only found in inactive biopedturbations and contained a greater seedbank in both active and inactive biopedturbations. *Cryptantha limensis* besides having a greater seedbank in BSC it showed a higher plant density in the biopedturbations. *Solanum phyllantum* and *Calandrinia alba* had a greater seed bank density in inactive biopedturbations and consequently showed a higher density

5   when emerge from inactive biopedturbations (Table 2). We hypothesized that some plant species benefit from the higher moisture content in the active biopedturbations, and some may also be favored by other characteristics of the biopedturbations. Nevertheless, we show the positive species-specific effect of biopedturbation on plant, which contributes to the plant community composition of the area.

10   **Fossorial birds as ecosystem engineers: interactions and impacts on plant communities**

The three components of the ecosystem studied have a series of interactions, thus we cannot cover them all we took a glance to the complex system. Fossorial birds destroy a small portion of BSC to create their burrows, and at the same time create an area where biological soil crust is buried under a mound of sand, which we called biopedturbations. This burial causes a negative effect on the biocrust organisms by stressing them (Rao et al., 2012). The full net of interaction between biological

15   soil crust and vegetation is highly complex and we cannot explain it completely. We based many of our assumptions on literature to reduce the complexity. We understand that BSC can have a positive effect (Belnap and Büdel, 2016; Boeken, 2008; Hawkes, 2004) and at the same time a negative effect on seed germination and establishment (Booth, 1941; Brotherson and Ruthforth, 1983; George et al., 2003; Johansen, 1993; Prasse and Bornkamm, 2000; Zhang et al., 2016, 2010). At the same time, vegetation could also provide a positive effect to the BSC in our system, by increasing soil organic

20   matter and soil fertility (Bowker, 2007; Li et al., 2007; Maestre et al., 2010). At last, the biopedturbations made by fossorial birds create small patches in the landscape where soil moisture was enhanced, soil nutrients were not altered and a disturbed surface was present, in contrast with the BSC matrix, in which resulted in more seedling germination and higher plant diversity in the transition from dry to wet season.

Fossorial birds cause biopedturbations that affect ecosystem functions, and can therefore be considered ecosystem engineers

25   (Jones et al., 1994). This is consistent with the intermediate disturbance hypothesis (Connell, 1978), where moderate disturbances maximize the species diversity in the system, and increase the special and temporal heterogeneity important for the maintenance of ecosystem biodiversity, structure and functioning (Huston, 1994; Pickett and White, 1985)

Data availability. The data is available in the supplement of this article.

30   **Author contribution.** MR and CA designed the research. MR collected samples from the field and did all the measurements. MR and CA analyzed the data.  MR prepared the manuscript with corrections from CA.

**Competing interests**. The authors declare that they have no conflict of interest.

**Special issue statement**. This article is part of the special issue "Biological soil crust and their role in biogeochemical processes and the cycling". It is not associated with a conference.

35   **Acknowledges.** We are grateful to Mery Suni, Tomás A. Carlo and friends of the Ecology Department of the Natural History Museum for the invaluable help and support in the development of this work. We also thank the Natural History Museum of Lima and the heads of the National Reserve of Lachay for the institutional support. Our gratitude extends to Anita Antoninka for her valuable help and support through the manuscript corrections. Furthermore, the present work was part of the undergraduate thesis of Maria C. Rengifo, and was possible thanks to the financial support of the Universidad Nacional

Mayor de San Marcos through the Vicerectorado de Investigación and the Instituto de Ciencias Biológicas Antonio Raimondi.

**Tables**

**Table 1: Chemical characteristics of the soil treatments. Different letters show statistical significance.**

| Treatment | Biopedtutbation soil | | Underlying soil | | BSC | | $P$-value |
|---|---|---|---|---|---|---|---|
| | Mean | SEM | Mean | SEM | Mean | SEM | |
| EC dS/m (1:1) | 1.99[a] | ± 0.50 | 0.67[b] | ± 0.34 | 0.40[b] | ± 0.08 | 0.043* |
| CaCO$_3$ % | 4.67[a] | ± 0.67 | 2.02[b] | ± 0.39 | 2.45[b] | ± 0.31 | 0.043* |
| pH (1:1) | 8.53 | ± 0.15 | 8.53 | ± 0.44 | 8.27 | ± 0.08 | 0.165 |
| OM% | 0.27[a] | ± 0.13 | 0.38[a] | ± 0.05 | 0.67[b] | ± 0.08 | 0.043* |
| K (ppm) | 242.00 | ± 6.00 | 379.33 | ± 13.61 | 428.67 | ± 99.93 | 0.165 |

| P (ppm) | 30.30 | ± | 6.91 | 22.53 | ± | 2.94 | 30.77 | ± | 2.17 | 0.165 |
|---------|-------|---|------|-------|---|------|-------|---|------|-------|

(*) Significant values, p<0.05. Median test for two independent medians.

25

**Table 2: Plant density (ind/m² ) of paired soil treatments in seedbank germination experiment and seedlings emergence observations.**

| Species | Seedbank germination experiment | | | | Seedlings emergence observations | | | |
|---------|------|--------|------|----------|------|--------|------|----------|
|  | BSC | Active | BSC | Inactive | BSC | Active | BSC | Inactive |
| *Cistanthe paniculata* | 2311.1 | 603.7 | 2408.8 | 1564.7 | 67.4 | 118.8 | 52.8 | 50.0 |
| *Fuertesimalva peruviana* | 137.0 | 318.5 | 229.4 | 155.9 | 23.6 | 88.2 | 21.5 | 72.2 |
| *Chenopodium petiolare* | 25.9 | 18.5 | 394.1 | 70.6 | - | - | - | - |
| *Cryptantha limensis* | 0.0 | 14.8 | 147.1 | 61.8 | 0.7 | 2.1 | 0.0 | 6.3 |
| *Calandrinia alba* | 3.7 | 3.7 | 79.4 | 91.2 | 0.0 | 0.7 | 0.0 | 2.1 |
| *Nolana humifusa* | 103.7 | 18.5 | 20.6 | 38.2 | 0.0 | 2.1 | 0.0 | 4.2 |
| *Solanum phyllanthum* | 0.0 | 3.7 | 2.9 | 79.4 | 0.0 | 0.7 | 0.0 | 3.5 |

| | | | | | | | | |
|---|---|---|---|---|---|---|---|---|
| *Oxalis lomana* | 81.5 | 14.8 | 0.0 | 5.9 | - | - | - | - |
| *Exodeconus prostratus* | 0.0 | 14.8 | 5.9 | 11.8 | 0.0 | 0.0 | 0.0 | 0.7 |
| *Palaua rhombifolia* | 11.1 | 0.0 | 14.7 | 14.7 | 0.0 | 0.7 | 0.0 | 0.7 |
| *Fuertesimalva limensis* | 37.0 | 3.7 | 2.9 | 0.0 | - | - | - | - |
| *Stenomesson coccineum* | 3.3 | 0.0 | 2.9 | | - | | . | . |
| *Solanum multifidum* | 1.6 | 0.0 | 0.0 | | - | - | - | |

**Figures**

[Figure]

Figure 1: Study area in the Coastal desert of Peru.

[Figure]

Figure 2. **A.** Landscape of the study area. The lower site of the hills, about 200 m above sea level, has a total surface cover of BSC except where it is disturbed by fossorial birds' burrows (Biopedturbation labeled in the picture). **B.** Biopedturbation studied, a 30x30 cm plot in the mound placed to observe seedling emergence. **C.** Burrowing owl *Athene cunicularia* standing on its biopedturbation. **D.** *Cistanthe paniculata* seedlings growing on BSC. **E.** Dark cyanobacterial biological soil crust that covers the study site.

[Figure]

Figure 3. **A.** Soil moisture content experimental design. Three plots were stablished in active and inactive biopedturbations: Scalped soil, Biopedturbation and BSC plot. **B.** Four soil surfaces were sampled, each with a particular soil profile.

[Figure]

Figure 4: Soil moisture content (%) of the two types of biopedturbations (Median ± 95% intervals). Different letters indicate significant differences (Wilcoxon signed rank test; P< 0.05)

[Figure]

Figure 5: Seed bank and field seedling emergence. Abundance, diversity Fisher alpha index and rarefaction curve of paired treatments of biopedturbation and BSC. Different superscripts indicate a significant difference between BSC and active or inactive disturbance at $P < 0.05$. AB: Active Biopedturbation. IB: Inactive Biopedturbation.

---

## Author Comment (AC2) · 19 Dec 2017

We thank gratefully the comments given, which we found constructive and improved tremendously the quality of this manuscript. We agree with most of the comments. We have revised the manuscript in the light of the comments. Below the separated specific comments we indicate our responses and we attached a new version of the manuscript in the supplement.

Comments by G. Kidron

The MS describes the effect of biopedoturbation on species diversity and plant germi-

nation in the Peruvian Desert. By disturbing the surface, fossorial birds create micro habitats that affect plant diversity and density. The increased diversity with increased heterogeneity is rather expected and the current MS adds additional information to a relatively large bulk of literature that exists on this topic. Nevertheless data from the Peruvian Desert is an important contribution. However, unfortunately, the MS is premature. While the authors describe the changes in seed bank and species diversity, no satisfactory explanation for this phenomenon is provided. This is a major obstacle once publication in a leading journal is sought. The MS suffers from additional drawbacks (lack of data, unclear Methodology), while the structure, flow and choice of citations also need major improvements.

Main points 1. The topic presented is not new and adds to many other publications on biopedoturbation, as thoroughly summarized by Whitford and Kay (1999). Whereas the authors try to link the findings to the presence of biological soil crusts (BSCs), the data presented are fragmentary and not convincing. The authors suggest that higher moisture availability at the mounds and lack of runoff there may explain the higher diversity and biomass at the mounds. Yet, the authors (1) present only three dates with moisture data throughout the entire growing period (2) the authors assume a linear increase in moisture from day 5 to day 60 (Fig. 2) although intensive fluctuation in moisture is expected due to the erratic nature of precipitation in deserts (3) no rain data are provided, which does not allow for a proper evaluation of the data (4) the values used for the moisture are not clear (gravimetric? volumetric? ratio of WHC?).

RE: We apologize for the mistake made regarding the values for the moisture content. We used the gravimetric moisture content, and the formula was added to the Method section. The Lomas of central coast of Peru have a well-marked seasonality, with very small amounts of precipitations in the form of fog, and we don't consider them to have an erratic nature of precipitation. We add a reference values of the precipitation to the area to make a better understanding of the climate. Since our objective was not to show changes in moisture through time but among treatments, and addressing a

referee suggestion, we changed the moisture analysis and only used the data set of the last measurements, were the three surface were comparable.

2. The data should include a detailed account of the research site (general description of the geomorphology and/or dunes; the particle size distribution, i.e., the amount of sand, silt and clay; the main microorganisms within the BSC and possible their chlorophyll content in mg/m2 as well as plant cover) and description of the disturbance (are the three birds mentioned have the same disturbance? What is the size of the mounds?). It should be accompanied by photographs that show the research site with the BSCs, the mounds created by the birds and photographs from the experimental design. Longterm precipitation at the site, including the possible contribution of fog and dew (approximate precipitation) should be added.

RE: We add a more extend description of the area, and remarked the gap in scientific literature on biological soil crust in the area. We add the description of the birds' disturbances and range areas of the sized, but not much has been reported about density or longevity. We add photographs of the area, the mounds, the plots and the biocrust.

In addition, clear hypotheses should be outlined and the rationale for measuring each of the variables should be thoroughly explained (for instance, what is the rationale of the chemical analysis of the crust? of measuring the calcium carbonate or EC?).

RE: We restructure the Introduction and state clearly the hypotheses. We also added a brief explanation of the chemical properties measure in the soil.

The Methodology should be thoroughly explained (statistics included). For instance, how do the authors define and differentiate between active and inactive mounds? Also, the methods or devices used for measuring each variable should be indicated, as well as the nutrient species. For instance: did the authors examine total or available P?

RE: Done. We explain thoroughly the methodology, add a graphic to visually help understand the experimental design for the moisture measurements and add a definition

of active and inactive biopedturbations.

3. The Ms structure. Generally, the flow should be substantially improved. In essence the MS lacks Introduction. The Introduction should include general theories regarding the effects of disturbances on the ecosystem, with a specific emphasis on deserts and BSCs, especially on sand-covered BSCs.

RE: We restructure and improved the introduction.

The Discussion should focus on the findings, discussing the similarities/differences with previous publications and the implications for the ecosystem. For instance, it is generally assumed that water availability at the mounds is lower (Moorhead et al., 1985), in contrast to the authors' conclusion. This should be thoroughly discussed.

RE: We improved the discussion section and add the findings of Boeken and Shachak (1994) that also found mounds to be dryer than the biocrust.

Also, the analysis is not clear. For instance, there are two main variables that may negatively affect moisture: loss of water due to runoff or increased evaporation. Both possibilities should be discussed.

RE: We briefly address how both mechanisms could be possible regarding our findings.

4. The choice of references is unclear. The link between the mentioned topic and the references should be improved. Reports and abstracts should be avoided unless no other material exists. Book chapters and review papers should at best accompany peer review journal articles with empirical data (rather than being used as central references).

RE: We improved some of the references. Although information of the local environment is limited in scientific literature we still need to cite reports.

Grouping together many topics (6th and 7th line in the Introduction) cannot guide the reader. Citations that refer to trivial points should be eliminated.

RE: We restructure the Introduction

5. Many of statements do not reflect the state-of-the-art knowledge and the picture that emerges is rather simplistic. For instance, do crusts necessarily promote plant survival (section 1 of Introduction)? Are BSCs necessarily hydrophobic (upper p. 6)? Will buried crust 'stop' infiltration (upper p. 6)? Do BSCs loose their water following the consumption of water by the microorganisms?

RE: We improved the discussion to address those issues.

Please also note the supplement to this comment:
https://www.biogeosciences-discuss.net/bg-2017-376/bg-2017-376-AC2-supplement.pdf

---

## Author Comment (AC3) · 19 Dec 2017

We thank gratefully the comments given, which we found constructive and improved tremendously the quality of this manuscript. We agree with most of the comments. We have revised the manuscript in the light of the comments. Below the separated specific comments we indicate our responses and we attached a new version of the manuscript in the supplement.

Comments by S. Chamizo

This manuscript analyses the effect of biopedturbations on soil properties and plant

abundance and diversity compared to BSCs, which is a very uncommon topic in the scientific literature regarding BSC and which makes the content of the manuscript interesting and novel. However, I have some concerns about the design of the experiment and discussion of the results obtained, especially regarding the effect of BSC on plant diversity and abundance.

I have three main points to highlight: First, different plots have been set up and samples have been collected from different places for soil moisture, chemical properties and seedling analyses, so it makes difficult to establish direct relationships between soil properties and abundance and diversity of plants in BSCs and biopedturbations. Nevertheless, if results about soil properties and seed bank and seedlings are presented, authors should make an effort to discuss these results in an integrated way, trying to link, to some extent, the effects on soil moisture and chemical properties in BSCs and biopedturbations with the results obtained regarding plant abundance and diversity. As written now, the Discussion looks like different paragraphs addressing independent results and without linking one result with others.

RE: We agree and improved the Discussion to create a better linked between the results.

Second, for the chemical properties, seed bank and seedling emergence experiments (it is not clear to me if also for the soil moisture), it has been compared BSC and the removed soil by biopedturbation (which is on top of a BSC), but not undisturbed soil devoid of BSC (or bare soil). This is important to really understand the effect of BSC on soil properties and plant establishment, as compared to bare soils.

RE: We agree, but it was not possible to compare BSC to actual bare soil, because in the study area the biological soil crust has a total cover of the soil surface. Which greatly differs from other parts of the world where biocrust grows in the interspaces. We add a better description of the area to explain this issue.

Third, it is said that BSCs have a negative effect on plants, but this conclusion is not

clear to me from the experiments conducted and the results obtained as, on one hand, there is no comparison of the BSC with bare soil to clearly understand the effect of the BSC and, on the other hand, the disturbed soil (by biopedturbation) lies on the BSC and thus, the BSC might have indirect effects on seedling emergence by contributing with longer moisture retention and higher nutrient release to the mound of sand. The authors should discuss these points adequately in the manuscript.

RE: We briefly increase the discussion on this issue and remarked that we are basing many of our assumptions in literature only, since no further studies had been done in biocrust in this ecosystem.

As a general comment, the language of the manuscript should be thoroughly revised by an English native speaker.

RE: Done

More detailed comments are: Page 4, MM. Were samples for soil moisture, chemical properties and seedlings determination taken in areas next to each other? RE: We add the extension of the sampled area. All measurements and sampling were done in the 1.4 square km. Nevertheless, because we target biopedturbations in the area the sampling depends on the distribution of the biopedturbations, and because of that we used paired sampling in each section of the methods.

Also, indicate the period in which soil sampling was done (dry or wet season). Page 4, P10: What does "experimental plot" (after "the bare soil plot") mean? Please, explain the meaning of "active" and "inactive" biopedturbations.

RE: Done. We add the definition of active and inactive biopedturbations both in the Introduction and in the third paragraph of the 2.1 Method section.

Page 4, P15. It is said that soil moisture was measured at three times, but what do days 0, 5 and 60 represent? If they are three independent measurements at three different times, it should be represented as time 1, time 2 and time 3 (or by the date) but not as

a cumulative time since an initial time. In addition, the sentence "At day 0 the bare soil plot was sample with the undisturbed BSC layer, and immediately after the collection of the sample the BSC was removed" is not understandable. Was the bare soil plot soil devoid of BSC or soil with BSC in which the crust was removed? In the latter case, it is not measured soil moisture content in bare soil but in the soil underneath the crust. The soil below the BSC usually has better properties (higher EPS, N, aggregation: : :) than the bare soil and thus, soil moisture is likely higher in the soil beneath the crust than in adjacent soils devoid of BSC. If the BSC was removed from the soil and water content was measured in the underlying soil in day 0, what was measured after 5 and 60 days? Soil moisture in the scalped soil? Both measurements are not comparable because in day 0 the presence of the BSC conditions soil moisture in the underlying soil, while in the resting days, soil water content is measured in soil lacking the BSC. I think authors should show soil moisture data only for the days in which similar surface types are compared, and in the case of the bare soil, let clear that it consists of scalped soil where the top BSC was removed (which is not the same than bare soil).

RE: We addressed all of this points and (1) Reanalyzed the data and only consider the data set on the last moment in time, where all the surfaces were comparable, and (2) we changed the term 'bare soil' to 'scalped soil' because it fits the actual nature of the surface, as suggested.

Regarding the method, if moisture content is determined by weight, it is gravimetric water content (g H2O/kg soil), not volumetric water content.

RE: We apologize for the mistake made. We correct this to show the gravimetric water content in percentage, and we add the formula in the Methods.

Page 4, P20. Which soil depth was sampled for the analysis of chemical properties? Besides, the method used for the determination of each soil property should be explained. We add the depth of the soil sample for the chemical properties analysis. We specify that ". The routine soil analysis included the available phosphorus (P), exchangeable potassium (K), calcium carbonate (CaCO3), soil organic matter (O.M.), pH and electrical conductivity (E.C.)".

RE: We didn't find necessary to explain the method used for each chemical property, but we could add it briefly if it is need for a better construction of the article.

Page 4, P25. Please, homogenize the terminology for biopedturbation samples as different terms are used along the text ("biopedturbation plot", "removed soil from the loose soil of the entrance of the burrows", "disturbed soil").

RE: Done

Page 5. It should be included a section of "data treatment" or "statistical analyses" to explain how statistical differences were analysed and also to explain the indices of plant abundance or diversity used. Done

Page 5. In general, description of the results is very poor and should be greatly improved. Authors should describe more in detail differences in the properties analysed between BSC and biopedturbations.

RE: We improved the Result section

Page 5, P5. Please, describe first differences in soil moisture among times (also explains what the different times represent), and then, differences among "bare soil" (see my comment above), BSC and types of biopedturbations. We eliminate the measurements of the first to moments in time. And in the results we first compare active biopedturbations with the 2 treatments and then did the same for the inactive biopedturbations Page 5, P10. Results of chemical properties should be better described by comparing the BSC with the underlying soil and both with the disturbed soil for all soil properties. For instance, it could be shown average values of the different properties in BSCs compared to disturbed soil,

RE: We improved the results of chemical properties and add some values.

Pages 5-6. The Discussion should be substantially improved. Authors should make an effort to connect the different results obtained and, for instance, try to link the results of seed bank and seedlings with the results of soil moisture, organic matter and nutrients in BSC and biopedturbations.

RE: We separated the Discussion in sections, linked them, and extend the final paragraph to summarize and linked our conclusions

The manuscript should be also improved by comparing with other published studies that analyse the effect of biopertudbations on seedlings and by adequately explaining and discussing the positive and negative effects of BSC on seedling and plant establishment, and relating these effects with their effects on soil properties. I also recommend using more recent references in the Discussion as some of them are old and there is a large budget of articles recently published about the influence of BSC on soil properties (water content, nutrients).

RE: Done, we add literature about the effects of BSC on vegetation.

Page 6, P5-10. In this paragraph it is said that in the biopedturbation, the mound of sand is on top of the BSC. If the BSC acts as a seal on the soil surface limiting water infiltration into deeper soil, it could have a positive effect retaining moisture at the surface and keeping moisture longer in the sand above it, indirectly favouring seedling in the mound of sand. Page 6. Soil moisture, chemical properties and seedlings in biopedturbations and BSC are discussed separately, and no relationships and interactions between these properties have been discussed. For instance, higher organic matter and nutrients in BSC could be the reason for higher plant abundance. In contrast, lower moisture could be the reason for lower diversity, as only certain species better adapted to drier conditions could be competitive for growing in soils covered by BSCs, while others with more water requirements would grow better in biopedturbation-disturbed soils.

RE: We made a better linked between the parts of the study. It's hard to address the

last point in the discussion because all 13 species are native and well adapted to the environment

Page 6, P25-40- An important point to discuss is the different results found in the seedling greenhouse experiment and the field experiment. Such differences could be related to differences in water availability between both experiments that could strongly condition species diversity and abundance in the BSC under greenhouse and field conditions. In the greenhouse, samples were irrigated frequently and in this case, higher abundance of the seed bank was found in the BSC compared to biopedturbations, while in the field, with limited water availability, opposite results were found. Water, thus, appears to be a major driver for seedling abundance. This should be discussed in the Discussion.

RE: Done. We expand the discussion.

Page 6, P30-35. Together with moisture availability, I really think that the reduction in seedling emergence in BSC is greatly associated to a physical impediment: the seal created by the crust impedes seed penetration and leaves the seed more exposed (and less protected) to hostile environmental factors, at the same time that facilitates seed removal by wind. Page 7, P5-10. This paragraph is confusing and mix different ideas about BSC and plant interactions. The authors should explain along the Discussion the contrasting effects of BSC on vegetation, and why they can have positive and negative effects on vegetation. The sentence "At the same time, vegetation provides a positive effect to the BSC (Bowker, 2007), and because photosynthetic organisms compete to each other for resources, a negative effect is also expected" is not understandable and contradictory as it suggests a simultaneous positive and negative effect of vegetation on BSC. I do not think plants and BSC compete for water and nutrient resources, but that BSCs grow in the areas where water and nutrients are not available enough to allow plant establishment.

RE: We improved the discussion. We eliminate the statement that 'photosynthetic

organisms compete to each other for resources'

The sentence "and at a landscape scale the presence of ecosystem engineer would result in an increase of the species richness, along with the Competitive exclusion principle of Gause (Palmer, 1994) coexistence is allowed, and as a result vegetation increases its abundance and richness in an indirect way." is very abstract and not understandable in this context. Please, either rewrite it or delete this sentence.

RE: Done, we delete the sentence.

Page 7, P20. What do the authors mean by "relationships of a high order interaction"? It is not clear that BSCs have a negative effect on the plant community and that "biopedturbations attenuate the negative effect of BSC to the plant community". Likely, BSCs could have an indirect effect on the disturbed soil by maintaining soil moisture longer and by contributing nutrients to the mound of sands.

RE: We deleted the sentences involving the high order interaction, since our findings are not strong enough to support it.

In addition, it has not been analysed seedling abundance and diversity in BSCs compared to bare soil.

RE: This was not possible in our study area, because areas without biocrust are very distant. We add a better description of the study area to remarked the full cover of biocrust in the landscape

Some editing comments: Is the term "biopedturbation" more commonly used than "bioturbation"? The second one is more familiar to me.

RE: We used the term biopedturbation as a more specific term for the disturbances of animals to soil.

Page 5, P15. ": :where the BSC samples had a higher abundance of germinated seeds than active and inactive biopedturbations." Figure 2. Include the units for soil moisture

in axis Y. In addition, units in the legend seem to be wrong (gH2O/kg soil, not

RE: Our soil moisture data is in percentage as shows in the formula used.

Maria Cristina Rengifo

Please also note the supplement to this comment:
https://www.biogeosciences-discuss.net/bg-2017-376/bg-2017-376-AC3-supplement.pdf

**Supplement:**

**Disturbances of Biological Soil Crust by fossorial birds increase plant diversity in a Peruvian desert**

María Cristina Rengifo[1,3], Cesar Arana[1,2]

[1]Departamento de Ecología, Museo de Historia Natural de la Universidad Nacional Mayor de San Marcos, Lima, 15072, Perú.
[2]Laboratorio de Ecología y Biogeografía Terrestre de la Facultad de Ciencias Biológicas de la UNMSM, Lima, 15081, Perú.
[3]School of Forestry, Northern Arizona University, 200 E. Pine Knoll Drive, Flagstaff, AZ 86011, USA.

*Correspondence to*: Maria C. Rengifo (mcr335@nau.edu)

**Abstract.** The Lomas Formation are fog-dependent oases within the hyper arid band of the Peruvian coast. Biological soil crusts (BSC) form in the Lomas and interact with their fauna and flora. Here we asked if natural disturbances – biopedturbations - made by fossorial birds have an effect on seedling emergence in the Lomas Formations in the National Reserve of Lachay in Lima, Peru. We analysed active and inactive avian biopedturbations, undisturbed BSC and scalped soil field samples for moisture content, soil chemical properties and the seedbank and the field emergence of seedlings. Active biopedturbations had the highest soil moisture content and BSC showed the lowest values. Organic matter content was significantly higher in the BSC than the soil beneath it and the bare soil. However, $CaCO_3$ content and EC were higher in bare soil than the other treatments, and no significant differences were found in soil pH, phosphorus or potassium content between treatments. In the seedbank experiment, 13 herbaceous plant species were found; furthermore, biopedturbations had a higher diversity but lower abundance than the BSC. However, in the field observations biopedturbations had a higher diversity and abundance of seedlings than BSC and only 8 herbaceous species were found. The species *Fuertesimalva peruviana* (L.) Fryxell, *Exodeconus prostratus* (L'Hér.) Raf., *Cryptantha granulosa* I.M. Johnst. *Solanum phyllanthum* Cav. and *Calandrinia alba* (Ruiz & Pav.) D.C increased their abundance in biopedturbations. Our results showed the positive effects on seed germination and diversity of vascular plants by the natural disturbances made by fossorial birds in a unique ecosystem of the Peruvian desert, and demonstrates the importance of spatial and temporal heterogeneity for ecosystem structure and functioning.

**1 Introduction**

Natural disturbances and their impact in ecological processes has been broadly studied in drylands, and focused extensively on small mammals (Eldridge et al., 2012; Hobbs and Huenneke, 1992; Kerley et al., 2004; Schooley et al., 2000; Whitford and Kay, 1999). Soil disturbances made by burrowing animals directly modify habitats and modulate the availability of resources;
5   by which they are consider ecosystem engineers (Guo, 1996; Hansell, 1993; Jones et al., 1994; Wright et al., 2004). Although it's known that burrowing mammals contribute to the heterogeneity that support different plant communities at small and large scales in landscapes (Eldridge et al., 2012; Whitford and Kay, 1999), very little is described on fossorials birds with similar behavior on drylands.

In the hyperarid system of the Sechura-Atacama Desert, there is a fog oasis known as *Lomas* (Fig 1; Rundel et al., 1991b).
10   Despite the extreme climatic conditions, the coastal hills in *Lomas* are high in biodiversity and endemism of plants and animals (Dillon and Rundel, 1990; Ferreyra, 1983; Pulido et al., 2007). The dry conditions limit the establishment of perennial plants and allow the development of annual vegetation only in the winter months when marine fog and fine drizzle provide water (Ferreyra, 1953). Common birds found in the central Peruvian *Lomas* generate soil disturbances –biopedturbations-. Miners *Geositta* spp. and the burrowing owl *Athene cunicularia* excavate tunnels and create notable mounds in the landscape. These
15   disturbances occur mostly in the low elevation areas of the coastal hills (between 150 and 400 meter) where vegetation cover is limited and the soil cover is dominated by biological soil crust (i.e., BSC) (Ferreyra, 1953). BSC in the central Peruvian *Lomas* has been recently reported for the first time (Arana et al., 2016). A knowledge gap regarding the basics of biocrust ecology, specifically basic structural and taxonomic characterization, nutrient fluxes and interactions with higher taxa, needs to be filled.

20   This soil community is known as important components of arid ecosystems (Johansen, 1993; Li, 2012; West, 1990). Biological soil crusts found around the world increase the fertility of soils and affect the physical properties of it by altering water infiltration, runoff, albedo, and temperature (Belnap et al., 2016; Bowker et al., 2010; Prasse and Bornkamm, 2000; Weber et al., 2016; West, 1990; Zaady and Shachak, 1994). As a result of those effects, the emergence and survival of vascular plants can be promoted (Jones et al., 1994, 1997).  Although effects on vascular plants are more variable and have negative effects
25   are also found depending on BSC and plant characteristics (Boeken and Shachak, 1994; Bowker, 2007; Li et al., 2010), which develops a greater complex scenario for biocrust and vascular plants interactions.

In order to create a basic understanding of the interactions between the components of the *Lomas* ecosystem, we look into the hyperarid system dominated by biocrust and the plant community response to biopedturbations. We hypothesized that the biopedturbations will create a positive effect on plant diversity and might be linked to some abiotic factors. The soil removal
30   and generation of mounds would increase soil moisture content and nutrient availability and increase seed germination. We created a study to specifically test the effects of fossorial birds' disturbances on soil moisture content, physical characteristics, seedbank germination and the emergence of annual plants. The study provides new information on the complexity and functioning of this understudied ecosystem and quantifies the relevance of interaction among its components.

In our study we targeted biopedturbations generated by fossorial birds that are the major disturbing agent in our site. The
35   landscape seen in the lowest part of the hills (*Lomas*) has a flat topography with a narrow inclination that gets steeper going east as we get closer to the top of the hills. The vast area is fully covered with biological soil crust, and only during 4 to 5 months of the wet season we see the establishment of the annual vascular plants. The rest of the year the area lacks higher plants cover. The birds' biopedturbations create bare patches observed with the naked eye across the landscape (Fig. 2). When burrows were active, we targeted the disturbed mounds of soil that have a larger area than the burrow entrance. The
40   soil profile of the mounds has 3 basic layers: an underlying sandy soil, a biocrust layer and another soil layer on top from the

burrowing activities of the birds. When burrows are abandoned, the lack of bird activity allows the biocrust organisms in the area to colonize the disturbed top soil, and a new layer is added to the soil profile; and we consider this as the inactive biopedturbations.

**2 Methods**

2.1 Study site and biopedturbations

The Atacama Desert is a coastal desert that extends 3500 km, from the region north of Trujillo near the Ecuadorian border of Peru, to central Chile (Rundel, 1978). This desert owes its aridity to the persistent temperature inversion associated with the cool north flowing Humboldt Current and the generally stable position of the strong Pacific anticyclone. The Andes Mountains prevent moisture from the east (Houston and Hartley, 2003). The National Reserve of Lachay is located 105 km north from the city of Lima, in the central coast of Peru (S11°23.6', W77°23'). The reserve contains a unique fog and mist-fed ecosystem called Lomas Formations within the hyper arid band of the Peruvian coast (Fig. 1). The landscape is characterized by small hills that create a smooth gradient from 150 m to 750 m of altitude, a mean annual precipitation in the open of 168 mm yr$^{-1}$. and in the high humidity season, from July until September, a dense fog comes from the sea adds moisture to the hills allowing the establishment of endemic flora (Rundel et al., 1991a).

The study area is located in the lower part of the hills (S11° 23.87', W77° 23.13') in approximately 1.4 km$^2$ with a smooth gradient from 150 to 250 m of altitude with a mostly flat topography (Fig. 2A). The seasonal vegetation found in the humid season is characterized by the presence of herbaceous plants of rapid flowering. The sandy loam soil is covered by a dark biological soil crust dominated by cyanobacteria (Arana et al., 2016), which is a type of BSC commonly found in warm deserts (Belnap et al., 2001; Pietrasiak, 2014). The BSC is 1-5 mm thick (Fig. 2E), and also has a low percentage (~25%) of moss (*Bryum argenteum* Hedw mostly) and some crustose and fruticose lichens. In this landscape three species of fossorial birds are the main agents of biopedturbations: the burrowing owl (*Athene cunicularia*), the coastal miner (*Geositta peruviana*) and the greyish miner (*Geositta maritima*).

For the purpose of this study we targeted the mound of removed soil that is placed over the surface as the biopedturbation (Fig. 2A-C). The birds break the BSC to create their burrows and the removed soil is placed on top of biocrust. We aimed for the average mounds that had an area of approximately 0.6 m$^2$; mostly from miners and some small ones from the burrowing owls. In the study area mounds of the two species of miners are indistinguishable one from another and range from 0.08 to 0.7 m$^2$. The mounds of the burrow owl are usually bigger and range between 0.4 to 1.3 m$^2$ (Unpublished data). We targeted active and inactive biopedturbations, the active ones were considered from the active burrows where the activity of the birds keep the top soil of the mound loose. Inactive biopedturbations were the mounds that were colonized by an early successional biocrust as a result of an abandoned burrow.

2.2 Moisture content

We examined the top 5 cm of soil moisture content between active and inactive biopedturbations, biological soil crust and scalped soil (Fig. 3). We established 26 experimental sites, 15 for active biopedturbations and 11 for inactive biopedturbations. Each experimental site consisted of three 30 x 30 cm plots placed not more than 1 m of separation: a scalped soil plot, a biopedturbation plot, and a BSC plot. The scalped soil plot was an 30x30 cm area where the surface layer

of BSC was removed to simulate bare soil surface. The biopedturbation plot was an area marked on the mound of sand, and the BSC plot was an area with undisturbed BSC. Selected biopedturbations were separated at least 10 m.

After two months of the BSC removal in the scalped soil plot, we sampled 100 g of the top 5 cm of soil inside each plot. Samples were collected in hermetic bags, and later in the laboratory were weighted, dried in an oven for 24 h at 105 °C, reweighed and their gravimetric moisture content calculated (Yair et al., 2011). Each sample represented a small fraction of the 0.09 m$^2$ area. The sampling collection was made at the end of October of 2015. This was an anomalous year influenced by the ENSO, 0 mm of precipitation was register in October, when 0.2 mm was expected.

$$Water\ content\ (\%) = \frac{w_w - w_d}{w_d} x\ 100$$

Where $W_w$ is the weight of wet soil in grams and $W_d$ is the weight of dry soil.

**2.3 Soil chemical properties**

To examine the soil chemical properties of the different layers of a biopedturbations we considered three soil treatments: (1) The undisturbed layer of BSC, (2) the underlying soil of 5 cm deep which was directly below the initially undisturbed layer of BSC sampled, and (3) the biopedturbation soil from the loose sand of the mound. Each treatment was replicated three times. The biopedturbations were separated by at least 10 m, and we used the undisturbed BSC next to each biopedturbation to diminish environmental heterogeneity. Every soil sample weighted approximately 500 g. The routine soil analysis included the available phosphorus (P), exchangeable potassium (K), calcium carbonate (CaCO$_3$), soil organic matter (O.M.), pH and electrical conductivity (E.C.), and was made by the Water, Soil and Environment Analysis Laboratory (LAASMA) of the Universidad Nacional Agraria La Molina, in Lima.

**2.4 Seedbank evaluation**

To evaluate the seed bank between biopedturbations and the soil covered with BSC, we took paired samples of soil from the mounds and from the undisturbed BSC next to the mound. Each soil sample was taken from a 10x10 cm area and 5cm deep; the paired samples were separated by approximately 1 m. We used 27 active biopedturbations and 34 inactive biopedturbations, each paired with an undisturbed BSC sample.

We conducted the experiment in a greenhouse at the UNMSM. The bagged soil samples were placed in plastic trays and arranged in a Latin Square design to eliminate positional effects within the greenhouse. The trays were watered regularly every two days with tap water, and the number of seedlings were recorded at frequent intervals. The germination beyond five weeks was found negligible and during the germination period there was no soil disturbances and seedling were removed only at the end of the experiment. The taxonomic determinations of species were made with dichotomous keys and specialized bibliography (Fryxell, 1996; Krapovickas, 2007; Lleellish et al., 2015; Sagástegui and Leiva, 1993; Tate, 2011).

**2.5 Field seedlings emergence**

We established a new set of plots in the field to observe the natural seedling emergence. We marked 30 x 30 cm paired plots, one on the biopedturbation and the other one in the undisturbed BSC, separated 1 m from each other. Paired sampling was need to diminish the environmental heterogeneity and compare plots with similar conditions due their proximity. We consider 15 replicates for active biopedturbations and 15 for inactive ones. In every plot we count and identify all the present seedlings. We evaluated the seedlings emergence at the beginning of the wet season in August 2016.

**2.6 Statistical analysis**

To determine if mounds, bare soil or BSC had different soil moisture, we used nonparametric related-samples Wilcoxon signed-rank test, to compare the median of paired samples. Soil chemical properties were compared in pairs with the nonparametric Independent-sample median test. Plant diversity was calculated with Fisher's alpha index defined by the formula S=a*ln(1+n/a) where S is number of taxa, n is number of individuals and a is the Fisher's alpha. We used PAST Paleontological Statistic software version 3.0 to extract the diversity index. Plant abundance values didn't go through any transformation. Plant diversity index and abundance values were analyzed by pairs using nonparametric related samples Wilcoxon signed-rank test. All the statistical nonparametric analyses were made in Software SPSS version 19, and α=0.05 for every case. Plant density was calculated by the sum of all seedling divide by number of samples and the area sampled in $m^2$.

**3 Results**

**3.1 Soil moisture content**

Soil moisture content showed different patterns among active and inactive biopedturbations. Active biopedturbations have the highest moisture content (0.71% ±0.164) among the soil treatments (Wilcoxon signed-rank test, p=0.041 paired to scalped soil, p=0.002 paired to BSC). On the other hand, inactive biopedturbations (0.66% ± 0.164) have similar moisture content than scalped soil (0.71% ± 0.152) and that BSC (0.58% ± 0.152). Biological soil crust showed to have the lowest moisture content in both types of biopedturbations (Wilcoxon signed-rank test, p<0.05 for active and inactive biopedturbations).

**3.2 Chemical properties**

Our results show that no significant difference in the potassium and phosphorous content among the BSC layer, the underlying soil or the biopedturbation soil (Independent-sample median test, p>0.05). The percentage of soil organic matter in the BSC layer (0.67%) was significantly higher than the underlying soil (0.38%) and the biopedturbation soil (0.27%), Independent-sample test p=0.043; and no difference was found between the organic matter values of the underlying soil and the biopedturbation soil. The pH values remained similar among the treatments. Calcium carbonate and electric conductivity values showed the same trend between treatments, where the highest values were found in the biopedturbation soil, and the BSC layer and the underlying soil had similar values (Table 1).

**3.3 Effects on plant germination**

We found 13 different native plant species germinated from the overall seedbank. Among treatments, active biopedturbations had a mean of 10.5 germinated seeds, significantly lower (Wilcoxon signed rank test, p =0.001) than the 27.4 germinated seeds in the paired BSC. Inactive biopedturbations had the same trend, with a mean of 21.2 germinated seeds, significantly lower (Wilcoxon signed-rank test, p=0.046) than the 33.1 germinated seeds of the paired BSC. (Fig 3). The Fisher's alpha diversity index of germinated species was 1.05, and significantly higher than the 0.76 of the paired BSC (Wilcoxon signed-rank test, p=0.035), but the same trend was not statistically supported for diversity in active biopedturbations. The number of species expected in the rarefaction curves was similar between treatments. Nevertheless, the BSC had a lower expected richness compared to their paired active biopedturbations (Fig. 5)

In field observations (Table 2) we recorded the emergence of only eight species at the beginning of the wet season. In contrast to the abundance found in the seedbank, the natural seedling emergence was higher in biopedturbations, active biopedturbations had a mean of 21.9 seedlings, significantly higher than the mean 11 seedlings in the paired BSC (Wilcoxon signed-rank test, p=0.011). Inactive biopedturbations showed the same trend, with a mean of 12.5 seedlings, significantly higher than the 8.2 seedling in BSC (Wilcoxon signed-rank test, p=0.020). The diversity of seedlings between treatments was not significantly different, although inactive biopedturbations had a slightly higher diversity index compared to the

paired BSC (Wilcoxon test, p=0.507). The number of expected species in the rarefaction curves are higher in both active and inactive biopedturbations compared to their paired BSC. Inactive biopedturbations shows the highest expected richness between treatments (Fig. 5)

The floristic composition (Table 2) shows that *Cistanthe paniculata* is the species with the highest density in every
5   treatment, except for the inactive biopedturbations in the field observations, where the species with the highest density is *Fuertesimalva peruviana*. We found species that have a higher density in inactive biopedtubations in both the seedbank and in the natural emergence: *Exodeconus prostratus, Solanum phyllanthum* and *Calandrinia alba*. An additional species, *Cryptantha limensis* also show a higher abundance in inactive biopedturbations in the field observations but without the same pattern in the seedbank.

10   **4 Discussion**

Our study is the first to look at how biopedturbation by fossorial birds alters the soil chemistry, moisture and potential for plant germination in the Lomas region of the Atacama Desert. This area is fully covered in dark cyanobacterial biocrusts except where burrowing activity has occurred. Burrowing fossorial birds are acting as ecosystem engineers, opening up niches for plant germination. Our work shows that biopedturbations had a positive effect on the plant community by
15   increasing the germinating plant diversity in the transition to wet season. Some annual plant species benefit from both active and inactive biopedturbations.

**Effects of biopedturbation on soil moisture**

Areas with BSC presented the lowest values of soil moisture content, that is a possible result of a low water infiltration, because the BSC tends to seal the soil surface (Brotherson and Ruthforth, 1983; Zhang et al., 2010). Factors like biological
20   soil crust characteristics, the topography and soil types (Chamizo et al., 2012) can cause BSC to increase (Brotherson and Ruthforth, 1983; Bu et al., 2015) or decrease (Eldridge et al., 2000; Gao et al., 2010; Kidron and Yair, 1997; Yair, 1990) infiltration rates in the soil. It was suggested that in sandy soils the BSC decrease the infiltration as a result from the reduction in the porosity (Warren, 2001). The biological soil crust of our study may decreased the water infiltration and as a consequence diminish the soil moisture content, this results also agreed with studies in the Negev desert (Eldridge et al.,
25   2000; Keck et al., 2016) which resembles physical characteristics of our studied area. Whereas the cyanobacteria and lichens of the BSC decrease the permeability of the soil (Loope and Gifford, 1972), the bare soil doesn't (Keck et al., 2016). Our data shows the same results, where the scalped soil had higher moisture content than BSC, that could be explained by the permeability, because biological soil crust tend to seal the soil surface (Booth, 1941; Brotherson and Ruthforth, 1983; George et al., 2003) and consume the water available in the most superficial soil layer (Bu et al., 2015; Gao et al., 2010)

30   On the other hand, biopedturbations present higher values of moisture content. In the case of active biopedturbations, moisture content was higher than the other soil surfaces, but the inactive biopedturbations show similar moisture content than the bare soil. We hypothesized that the soil profile of active biopedturbation (Fig. 3) allows water to infiltrate easily through the first layer of sand, and because of the hydrophobic characteristics and the water absorption of biocrust organisms (Kidron et al., 1999; Rodríguez-Caballero et al., 2013) in the buried BSC layer the moisture is better retained, compared to
35   the bare soil and the soil cover with BSC. Our results differed completely with a study in Northern Negev of Israel (Boeken and Shachak, 1994) that found that man-made mounds consistently presented the lowest moisture content compare to the BSC matrix, the main difference with this study is the characteristic of the man-made mounds, that were bigger and taller than the bird mounds. The soil moisture content on the first 15 cm was only loose soil, where in our study the first 5cm involves the BSC layer. The inactive biopedturbation, has an additional incipient BSC layer that could diminish the water
40   infiltration by reduction of porosity and the consumption of the water available in the most superficial soil layer and the (Bu et al., 2015; Gao et al., 2010; Warren, 2001), but not as much as a well develop BSC that shows the lowest values among treatments. As a result, the moisture content in inactive biopedturbations is similar to the scalped soil and to the BSC.

**Effects of biopedturbation on soil chemistry**

The soil chemical properties give us an approximation on the disturbance effect of the fossorial birds. Although many physical and chemical soil characteristics are insightful way to understand the ecological and biochemical processes occurring in the system, our study resources limited the extent of measurable characteristics. A basic routine soil analysis provided an insight of soil fertility and compared if the biopedturbations generate a great effect on those characteristics.

5    The nutrients potassium and phosphorus and the pH didn't show statistical differences between treatments, suggesting that the bird's biopedturbations do not alter significantly the nutrient soil content. Potassium values were slightly higher in the BSC, which is expected since it has been established before that BSC increases the soil fertility (Belnap and Harper, 1995; Harper and Pendleton, 1993). Though also bird droppings may contribute to soil fertility, our results did not support it. Differences in carbon or nitrogen content might be a better indicator of fertility, and thorough studies should be done in the
10   *Lomas* environment. Soil pH has been reported in other studies to not be significantly altered by the presence of BSC (Evans and Johansen, 1999; Guo et al., 2008; Kidron et al., 2015), and our results suggest the same. Soil organic matter was expected to be higher in the BSC in comparison with the underlying and biopedturbation soil, since is directly related to the organic carbon and BSC have high concentration of organism living in it (Delgado-Baquerizo et al., 2016; Guo et al., 2008). organic matter content, suggesting that neither the occasional bird droppings, nor the destruction and removal of the BSC
15   increases the organic matter in the other soil surfaces.

Unlike other studies (Guo et al., 2008), calcium carbonate content was not higher in the BSC, instead the biopedturbation soil had the higher values, this results may be explained by the abundant remains of gastropods shell's present in the locality, and not a reflection of increase nutrients in the soil. At the same time, since electric conductivity is influenced by the concentration of calcium carbonate BSC did not show high values of EC; despite being expected due to the concentration of
20   ions released by the decomposition of microorganisms. Although some differences in chemical properties where found and were expected, the lack of significant variation in K and P nutrients and pH between treatments shows that the biopedturbations might not have a greater effect on those soil characteristics due to the small area affected and may not have a great contribution in plant establishment.

**Biopedturbation effects on plant communities**

25   The plant community responded different in the greenhouse experiments than in the field emergence observations. Studies on BSC effects on vascular plants germination and establishment have contradictory results (Bowker, 2007) some consider that BSC can affect negatively plant density (Johansen, 1993; Prasse and Bornkamm, 2000), other showed the positive effect in semiarid ecosystems (Boeken et al., 2004; Defalco et al., 2001) and many species-specific effects (Hawkes, 2004; Su et al., 2009). By stop the limitation of water availability on the soil, the total germination of the seed bank in the greenhouse
30   represent the total viable seeds contained in the first 5cm of the soil (Thompson and Grime, 1979; Zhang et al., 2010). We found that biopedturbations are significantly less abundant in seeds of annual vascular plants in contrast with the BSC; nevertheless, are more diverse and rich in species composition. Even though a smooth biocrust increases the chances that seed may be taken by the wind (Belnap, 2006; Boeken and Shachak, 1994), plant litter annually accumulated on the soil surface remain along with a large seed load. The mounds of sand over the BSC do not present the same quantity of plant
35   litter and as a consequence it would be expected to have less seeds. A higher diversity in biopedturbations suggests a positive effect in the vegetation composition, but it would be necessary to study more thoroughly this phenomenon to understand if those biopedturbations work as a seed trap, if the higher diversity is product of the soil removal, or if the loose soil creates a better new topography compared to the tightly woven BSC surface (Boeken and Shachak, 1994).

Our field data taken in the transition from dry season to wet season shows that biopedturbations have higher abundance and
40   diversity in contrast with the BSC. Since the seedbank is more abundant in the BSC, the low emergence in the field compared to biopedturbations might be explained by the soil moisture content, that we found to be higher in the biopedturbations. And, although diversity was no significantly different, biopedturbations in the field show a higher species richness as seen in the seed bank.
Moreover, biopedturbations had a positive effect in some specific species: *Fuertesimalva peruviana,* in spite of been a high
45   density specie in the BSC, almost quadrupled its abundance in active and inactive biopedturbations in the field, and doubled

its seedbank density. *Exodeconus prostratus* the species with the lowest density in the field was only found in inactive biopedturbations and contained a greater seedbank in both active and inactive biopedturbations. *Cryptantha limensis* besides having a greater seedbank in BSC it showed a higher plant density in the biopedturbations. *Solanum phyllantum* and *Calandrinia alba* had a greater seed bank density in inactive biopedturbations and consequently showed a higher density

5   when emerge from inactive biopedturbations (Table 2). We hypothesized that some plant species benefit from the higher moisture content in the active biopedturbations, and some may also be favored by other characteristics of the biopedturbations. Nevertheless, we show the positive species-specific effect of biopedturbation on plant, which contributes to the plant community composition of the area.

10   **Fossorial birds as ecosystem engineers: interactions and impacts on plant communities**

The three components of the ecosystem studied have a series of interactions, thus we cannot cover them all we took a glance to the complex system. Fossorial birds destroy a small portion of BSC to create their burrows, and at the same time create an area where biological soil crust is buried under a mound of sand, which we called biopedturbations. This burial causes a negative effect on the biocrust organisms by stressing them (Rao et al., 2012). The full net of interaction between biological

15   soil crust and vegetation is highly complex and we cannot explain it completely. We based many of our assumptions on literature to reduce the complexity. We understand that BSC can have a positive effect (Belnap and Büdel, 2016; Boeken, 2008; Hawkes, 2004) and at the same time a negative effect on seed germination and establishment (Booth, 1941; Brotherson and Ruthforth, 1983; George et al., 2003; Johansen, 1993; Prasse and Bornkamm, 2000; Zhang et al., 2016, 2010). At the same time, vegetation could also provide a positive effect to the BSC in our system, by increasing soil organic

20   matter and soil fertility (Bowker, 2007; Li et al., 2007; Maestre et al., 2010). At last, the biopedturbations made by fossorial birds create small patches in the landscape where soil moisture was enhanced, soil nutrients were not altered and a disturbed surface was present, in contrast with the BSC matrix, in which resulted in more seedling germination and higher plant diversity in the transition from dry to wet season.

Fossorial birds cause biopedturbations that affect ecosystem functions, and can therefore be considered ecosystem engineers

25   (Jones et al., 1994). This is consistent with the intermediate disturbance hypothesis (Connell, 1978), where moderate disturbances maximize the species diversity in the system, and increase the special and temporal heterogeneity important for the maintenance of ecosystem biodiversity, structure and functioning (Huston, 1994; Pickett and White, 1985)

Data availability. The data is available in the supplement of this article.

30   **Author contribution.** MR and CA designed the research. MR collected samples from the field and did all the measurements. MR and CA analyzed the data.  MR prepared the manuscript with corrections from CA.

**Competing interests**. The authors declare that they have no conflict of interest.

**Special issue statement**. This article is part of the special issue "Biological soil crust and their role in biogeochemical processes and the cycling". It is not associated with a conference.

35   **Acknowledges.** We are grateful to Mery Suni, Tomás A. Carlo and friends of the Ecology Department of the Natural History Museum for the invaluable help and support in the development of this work. We also thank the Natural History Museum of Lima and the heads of the National Reserve of Lachay for the institutional support. Our gratitude extends to Anita Antoninka for her valuable help and support through the manuscript corrections. Furthermore, the present work was part of the undergraduate thesis of Maria C. Rengifo, and was possible thanks to the financial support of the Universidad Nacional

Mayor de San Marcos through the Vicerectorado de Investigación and the Instituto de Ciencias Biológicas Antonio Raimondi.

**Tables**

**Table 1: Chemical characteristics of the soil treatments. Different letters show statistical significance.**

| Treatment | Biopedtutbation soil | | Underlying soil | | BSC | | $P$-value |
|---|---|---|---|---|---|---|---|
| | Mean | SEM | Mean | SEM | Mean | SEM | |
| EC dS/m (1:1) | 1.99[a] | ± 0.50 | 0.67[b] | ± 0.34 | 0.40[b] | ± 0.08 | 0.043* |
| CaCO$_3$ % | 4.67[a] | ± 0.67 | 2.02[b] | ± 0.39 | 2.45[b] | ± 0.31 | 0.043* |
| pH (1:1) | 8.53 | ± 0.15 | 8.53 | ± 0.44 | 8.27 | ± 0.08 | 0.165 |
| OM% | 0.27[a] | ± 0.13 | 0.38[a] | ± 0.05 | 0.67[b] | ± 0.08 | 0.043* |
| K (ppm) | 242.00 | ± 6.00 | 379.33 | ± 13.61 | 428.67 | ± 99.93 | 0.165 |

| P (ppm) | 30.30 | ± | 6.91 | 22.53 | ± | 2.94 | 30.77 | ± | 2.17 | 0.165 |
|---------|-------|---|------|-------|---|------|-------|---|------|-------|

(*) Significant values, p<0.05. Median test for two independent medians.

25

**Table 2: Plant density (ind/m² ) of paired soil treatments in seedbank germination experiment and seedlings emergence observations.**

| Species | Seedbank germination experiment | | | | Seedlings emergence observations | | | |
|---------|------|--------|------|----------|------|--------|------|----------|
|  | BSC | Active | BSC | Inactive | BSC | Active | BSC | Inactive |
| *Cistanthe paniculata* | 2311.1 | 603.7 | 2408.8 | 1564.7 | 67.4 | 118.8 | 52.8 | 50.0 |
| *Fuertesimalva peruviana* | 137.0 | 318.5 | 229.4 | 155.9 | 23.6 | 88.2 | 21.5 | 72.2 |
| *Chenopodium petiolare* | 25.9 | 18.5 | 394.1 | 70.6 | - | - | - | - |
| *Cryptantha limensis* | 0.0 | 14.8 | 147.1 | 61.8 | 0.7 | 2.1 | 0.0 | 6.3 |
| *Calandrinia alba* | 3.7 | 3.7 | 79.4 | 91.2 | 0.0 | 0.7 | 0.0 | 2.1 |
| *Nolana humifusa* | 103.7 | 18.5 | 20.6 | 38.2 | 0.0 | 2.1 | 0.0 | 4.2 |
| *Solanum phyllanthum* | 0.0 | 3.7 | 2.9 | 79.4 | 0.0 | 0.7 | 0.0 | 3.5 |

| | | | | | | | | |
|---|---|---|---|---|---|---|---|---|
| *Oxalis lomana* | 81.5 | 14.8 | 0.0 | 5.9 | - | - | - | - |
| *Exodeconus prostratus* | 0.0 | 14.8 | 5.9 | 11.8 | 0.0 | 0.0 | 0.0 | 0.7 |
| *Palaua rhombifolia* | 11.1 | 0.0 | 14.7 | 14.7 | 0.0 | 0.7 | 0.0 | 0.7 |
| *Fuertesimalva limensis* | 37.0 | 3.7 | 2.9 | 0.0 | - | - | - | - |
| *Stenomesson coccineum* | 3.3 | 0.0 | 2.9 | | - | | . | . |
| *Solanum multifidum* | 1.6 | 0.0 | 0.0 | | - | - | - | |

**Figures**

[Figure]

Figure 1: Study area in the Coastal desert of Peru.

[Figure]

Figure 2. **A.** Landscape of the study area. The lower site of the hills, about 200 m above sea level, has a total surface cover of BSC except where it is disturbed by fossorial birds' burrows (Biopedturbation labeled in the picture). **B.** Biopedturbation studied, a 30x30 cm plot in the mound placed to observe seedling emergence. **C.** Burrowing owl *Athene cunicularia* standing on its biopedturbation. **D.** *Cistanthe paniculata* seedlings growing on BSC. **E.** Dark cyanobacterial biological soil crust that covers the study site.

[Figure]

Figure 3. **A.** Soil moisture content experimental design. Three plots were stablished in active and inactive biopedturbations: Scalped soil, Biopedturbation and BSC plot. **B.** Four soil surfaces were sampled, each with a particular soil profile.

[Figure]

Figure 4: Soil moisture content (%) of the two types of biopedturbations (Median ± 95% intervals). Different letters indicate significant differences (Wilcoxon signed rank test; P< 0.05)

[Figure]

Figure 5: Seed bank and field seedling emergence. Abundance, diversity Fisher alpha index and rarefaction curve of paired treatments of biopedturbation and BSC. Different superscripts indicate a significant difference between BSC and active or inactive disturbance at $P < 0.05$. AB: Active Biopedturbation. IB: Inactive Biopedturbation.